# TGM: A Modular and Efficient Library for Machine Learning on Temporal Graphs

**Jacob Chmura**[1,2,*], **Shenyang Huang**[1,2,3,*], **Tran Gia Bao Ngo**[5], **Ali Parviz**[1,4],
**Farimah Poursafaei**[1,2], **Jure Leskovec**[5], **Michael Bronstein**[3,6],
**Guillaume Rabusseau**[1,7,8], **Matthias Fey**[10], **Reihaneh Rabbany**[1,2,8]

[*] Equal Contribution, [1]Mila – Quebec AI Institute, [2]McGill University, [3]University of Oxford,
[4]New Jersey Institute of Technology, [5]University of Manitoba, [6]Stanford University,
[7]AITHYRA, [8]DIRO, Université de Montréal, [9]CIFAR AI Chair, [10]Kumo.AI
Corresponding authors: {`jacob.chmura,shenyang.huang`}`@mail.mcgill.ca`

## Abstract

Well-designed open-source software drives progress in Machine Learning (ML) research. While static graph ML enjoys mature frameworks like PyTorch Geometric and DGL, ML for temporal graphs (TG), networks that evolve over time, lacks comparable infrastructure. Existing TG libraries are often tailored to specific architectures, hindering support for diverse models in this rapidly evolving field. Additionally, the divide between continuous- and discrete-time dynamic graph methods (CTDG and DTDG) limits direct comparisons and idea transfer. To address these gaps, we introduce Temporal Graph Modelling (TGM), a research-oriented library for ML on temporal graphs, the first to unify CTDG and DTDG approaches. TGM offers first-class support for dynamic node features, time-granularity conversions, and native handling of link-, node-, and graph-level tasks. Empirically, TGM achieves an average 7.8× speedup across multiple models, datasets, and tasks compared to the widely used DyGLib, and an average 175× speedup on graph discretization relative to available implementations. Beyond efficiency, we show in our experiments how TGM unlocks entirely new research possibilities by enabling dynamic graph property prediction and time-driven training paradigms, opening the door to questions previously impractical to study.

Code: tgm-team/tgm     Documentation: tgm.readthedocs.io

## 1 Introduction and Motivation

Advances in machine learning are driven by open, easy-to-use libraries that let researchers focus on developing frontier architectures. For example, deep learning research was propelled by Caffe (Jia et al., 2014), TensorFlow (Abadi et al., 2016) and PyTorch (Paszke et al., 2019). Similarly, developments in graph machine learning (Kipf & Welling, 2016; Veličković et al., 2017; Dwivedi & Bresson, 2020; Rampášek et al., 2022) are accelerated by libraries such as PyG (Fey & Lenssen, 2019; Fey et al., 2025) and DGL (Wang et al., 2019). However, both PyG and DGL are designed for static graphs and cannot capture the temporal dynamics of networks, known as Temporal Graphs (TGs). Real-world examples include transaction (Shamsi et al., 2022), social (Huang et al., 2023a), trade (Poursafaei et al., 2022b), and communication networks (Yoon et al., 2020) among others.

Recently, Temporal Graph Learning (TGL) has emerged to capture both spatial and temporal dependencies in networks (Cornell et al., 2025; Cao et al., 2020; Han et al., 2014). The field has seen growth with high-impact, cross-domain applications, such as LinkedIn's LiGNN system (Borisyuk et al., 2024) for user recommendations and mobility modelling that informed COVID-19 policy decisions (Chang et al., 2021). Unlike static graph ML, TGL must treat time as a first-class signal, making timestamps central to modelling and data processing. Despite research progress, software infrastructure has not kept pace.

**Limitations of existing libraries.** Current TG libraries (Yu et al., 2023a; Rozemberczki et al., 2021) are narrow in scope: many implement only a single algorithm family (Wang & Mendis, 2024;

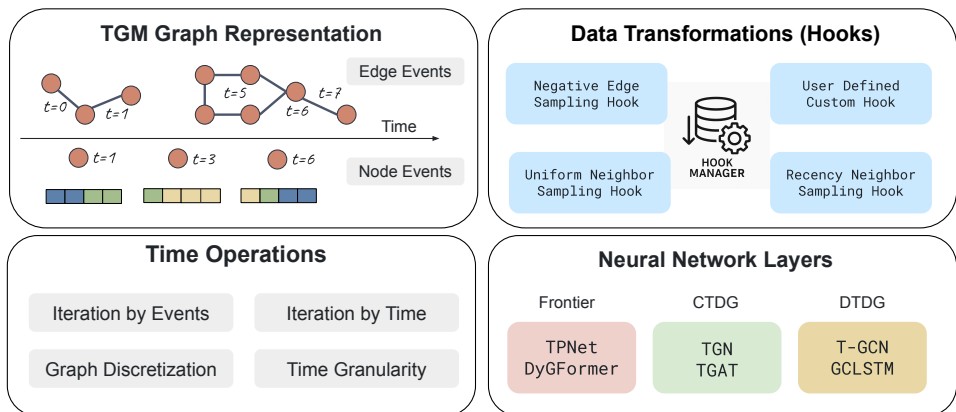

Figure 1: Overview of TGM features. TGM has native support for node events and unified continuous- and discrete-time graph iteration (left). Generic hooks formalize common TG transformations (top-right). TGM supports a broad range of temporal graph learning methods (bottom-right).

Zhou et al., 2023b) and most lack extensibility, resulting in a fragmented ecosystem. For instance, TGL (Zhou et al., 2022a), DistTGL (Zhou et al., 2023b) and TGLite (Wang & Mendis, 2024) are optimized for temporal message passing architectures (Rossi et al., 2020; da Xu et al., 2020) but do not support emerging transformer-based approaches (Yu et al., 2023a; Gao et al., 2025). Also, none provide time conversion operations which are critical for analyzing temporal granularity in TGs (Huang et al., 2024). Finally, existing libraries fall short on usability features needed to foster reproducible research such as profiling tools, test suites, and modular abstractions (see Table 1).

**Motivation for a unified framework.** Unlike NLP, where the transformer serves as a canonical architecture (Vaswani et al., 2017), TGL lacks a standard model family. This leads to fragmented and error-prone experimentation: continuous- and discrete-time models require entirely different data pipelines, while core operations such as temporal neighbor sampling and negative edge construction are implemented inconsistently. Without a unified framework, the community faces difficulties in fair benchmarking, rapid prototyping, and combining ideas across approaches.

**Our solution.** We introduce TGM, a modular and efficient framework for TGL research. TGM introduces several firsts: native support for node events, a generic hook mechanism that standardizes TG transformations, and unified support for both continuous- and discrete-time graphs, ending the long-standing separation between the two lines of research (Rossi et al., 2020; You et al., 2022). Node events naturally capture phenomena like social media posts or other user activity in real-world networks (Kazemi et al., 2020a). These abstractions unify diverse TG pipelines, lowering the barrier for practitioners and accelerating innovation. Beyond flexibility, TGM delivers efficiency: $7.8\times$ faster than DyGLib on standard TG models and an average speedup of $175\times$ on graph discretization.

In summary, the key properties of TGM are:

- **First unified library for TG.** TGM is the first library to support both continuous- and discrete-time graphs, treating them as distinct views of the same underlying data. We implement 8 methods from both CTDG and DTDG literature, including frontier models.
- **Time as a first-class citizen.** Time operations are central to TGs. TGM natively incorporates time granularity into its API, with built-in support for graph discretization and snapshot iteration.
- **Efficiency.** Our experiments show that TGM achieves an average $7.8\times$ faster end-to-end training than DyGLib, and $175\times$ faster graph discretization compared with existing implementations.
- **Research-oriented.** Designed for rapid prototyping, TGM emphasizes modularity and ease-of-use. Its novel hook mechanism standardizes temporal graph transformations while supporting the broadest range of TG tasks: link, node, and graph-level prediction.

## 2 RELATED WORK

**CTDG Methods.** Continuous-time Dynamic Graph (CTDG) methods process temporal graphs as streams of timestamped edge events. TGAT (da Xu et al., 2020) pioneered inductive representation learning on temporal graphs, and TGN (Rossi et al., 2020) generalized this approach into a widely adopted framework, with TGAT as a special case. Both rely on temporal neighbor sampling for message passing. More recently, DyGLib (Yu et al., 2023a) emerged as a popular library, introducing DyGFormer, one of the first transformer-based CTDG architectures inspired by their success in time series, NLP, and vision (Vaswani et al., 2017; Devlin et al., 2019; Dosovitskiy et al., 2021). Despite these advances, Poursafaei et al. (2022a) exposed flaws in prior evaluation and proposed EdgeBank, a strong heuristic baseline for link prediction. To address reproducibility, Huang et al. (2023a) introduced the large-scale Temporal Graph Benchmark (TGB), which we adopt for evaluating TGM. Recently, TPNet (Lu et al., 2024b) further advanced state-of-the-art link prediction by introducing temporal walk matrices with time decay, and is fully supported in TGM.

**DTDG Methods.** Discrete-time Dynamic Graph (DTDG) or snapshot-based methods represent temporal evolution as a sequence of static graph snapshots, adapting GNNs like GCN (Kipf & Welling, 2017) to this setting. GCLSTM (Chen et al., 2018) integrates GCNs with LSTMs (Hochreiter & Schmidhuber, 1997) to capture spatial and temporal dependencies, while PyG Temporal (Rozemberczki et al., 2021) provides a library of DTDG architectures for spatiotemporal graph learning. However, PyG Temporal lacks recent methods and standardized benchmarks like TGB. More recently, Unified Temporal Graph (UTG) (Huang et al., 2024) demonstrated a proof-of-concept for comparing CTDG and DTDG approaches via graph discretization. While UTG offers useful insights, its implementation is slow, limited to a few datasets, and not designed for reuse. In contrast, TGM supports fully vectorized graph discretization and time-iteration operations, unifying CTDG and DTDG within a single, robust framework and closing a long-standing gap in TGL.

**CTDG vs DTDG.** Discrete and continuous-time formulations differ primarily in their treatment of time: DTDGs aggregate changes into snapshots while CTDGs record each event, providing fine-grained temporal resolution (Kazemi et al., 2020b; Longa et al., 2023; Gravina & Bacciu, 2024). DTDGs offer simpler batch processing when data arrives regularly, whereas CTDGs are event-driven, making them better for capturing fine-grained dynamics and causal relationships. While DTDGs prioritize efficiency, they may lose important information when events occur more frequently than the snapshot rate (Kazemi et al., 2020b). Prior discussions largely remain conceptual and offer no practical framework for comparing or unifying the two paradigms. For example, applying DTDG models to CTDG tasks requires discretizing continuous-time events into snapshots.

**TGL Libraries.** Several libraries support temporal graph learning including DyGLib (Yu et al., 2023b), TGL (Zhou et al., 2022b), DistTGL (Zhou et al., 2023a), TGLite (Wang & Mendis, 2024), and TSL (Cini & Marisca, 2022). DyGLib provides pipelines for continuous-time models but is limited by scalability, lack of modularity, and weak support for discrete-time methods (Gastinger et al., 2024). TGL and DistTGL offer large-scale sampling and multi-GPU execution but lack a researcher-friendly interface and have seen few recent updates. TGLite focuses on continuous-time message-flow models, while TSL addresses spatiotemporal modelling on static graphs.

Table 1 summarizes key aspects of these libraries. TGM stands out as the only library that supports both CTDG and DTDG methods, bridging continuous- and discrete-time research paradigms. Its efficient and modular design facilitates flexible experimentation, while support for time conversion and dynamic node events enables diverse temporal graph learning tasks. Additionally, comprehensive tests and system profiling ensure reproducibility and provide research-ready infrastructure.

## 3 TGM FRAMEWORK

In this section, we present the foundation and concepts in TGM. In Section 3.1, we introduce the formulation of temporal graph in TGM and unify continuous- and discrete-time formulations by defining them as different ways of iteration over the graph. We also define the graph discretization operation as a principled way to map from continuous events to snapshots. Then in Section 3.2, we introduce the TGM hook formalism, a modular abstraction for composing graph operations. Together, these elements inform the software and system design in Section 4.

Table 1: Comparison of TGL libraries. TGM is the only library that meets all desirable criteria for TGL research while other libraries lack one or more criteria.

| Library | **TGL Features** | | | | **Software Infrastructure** | | | |
| | CTDG | DTDG | Time Ops. | Node Events | Modular | Efficient | Unit Tests | Profiling |
|---|---|---|---|---|---|---|---|---|
| TGM (ours) | ✓ | ✓ | ✓ | ✓ | ✓ | ✓ | ✓ | ✓ |
| DyGLib | ✓ | ✗ | ✗ | ✗ | ✗ | ✗ | ✗ | ✗ |
| TGL | ✓ | ✗ | ✗ | ✗ | ✗ | ✓ | ✗ | ✓ |
| TGLite | ✓ | ✗ | ✗ | ✗ | ✓ | ✓ | ✓ | ✓ |
| PyG Temporal | ✗ | ✓ | ✗ | ✗ | ✓ | ✓ | ✓ | ✗ |

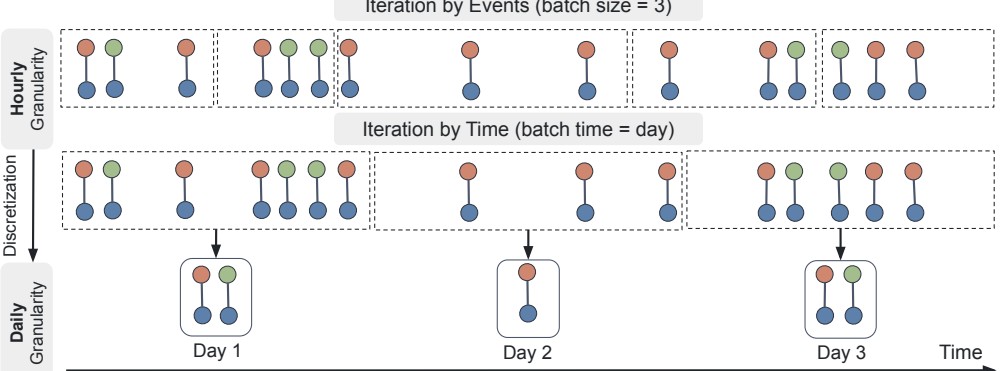

Figure 2: TGM supports iteration by events and time. Discretization maps fine-grained timestamps (e.g., hourly) to coarser timestamps (e.g., daily), aggregating duplicated edges in the process.

## 3.1 TGM TEMPORAL GRAPH FORMULATION

Here, we first introduce the notation of temporal graphs in TGM. On temporal graphs, events are considered as a fundamental unit for representing the network's evolution (Kazemi et al., 2020a). To capture changes in graph structure and features, TGM distinguish between the following event types:

**Definition 3.1** (Node and Edge Events). An edge event $(t, s, d, \mathbf{x}_{edge})$ is an interaction between two nodes $s$ and $d$ at time $t$ where $\mathbf{x}_{edge} \in \mathbb{R}^{d_{edge}}$ is the associated edge feature vector. A node event $(t, s, \mathbf{x}_{node})$ represents the arrival of new features $\mathbf{x}_{node} \in \mathbb{R}^{d_{node}}$ at node $s$ and timestamp $t$.

**Definition 3.2** (Temporal Graph). A temporal graph is a sequence of time-ordered events: $\mathcal{G} = \{e_0, ..., e_T\}$. Each event $e_i$ can be an edge event or a node event. Also, $\mathcal{G}$ can be associated with a static node feature matrix $\mathbf{X} \in \mathbb{R}^{n \times d_{static}}$ where $n$ is the number of unique nodes in $\mathcal{G}$. For any time interval $\mathcal{T} \subset \mathbb{R}^+$, the temporal sub-graph $\mathcal{G}|_{\mathcal{T}}$ contains all events in $\mathcal{G}$ intersecting $\mathcal{T}$.

**Representing Continuous and Discrete-Time Graphs.** As temporal graphs are represented as sequences of events in TGM, TGM doesn't treat CTDG and DTDG as different data types but rather as distinct ways of data iteration. We consider that any temporal graph admits a native time granularity $\tau$: the coarsest unit of time (e.g., seconds) that still discriminates between all event timestamps. If real-world time is unavailable (e.g., due to privacy), TGM employs a special event-ordered granularity $\tau_{\text{event}}$, preserving only the relative order of events but lacks correspondence to a real-world time granularity, thus $\tau_{\text{event}}$ is excluded from any time operations. Lastly, note that time granularities can be compared: $\hat{\tau} \leq \tau \iff \tau \text{ is coarser than } \hat{\tau}$. This view unifies CTDG and DTDG as alternative ways of iterating over the same event stream:

**Definition 3.3** (CTDG: Event-based iteration). A CTDG is often expressed as a stream of events (Kazemi et al., 2020a; Huang et al., 2023b; Rossi et al., 2020; You et al., 2022). In TGM, iterating a CTDG corresponds to using the event-ordered granularity $\tau_{\text{event}}$. Each batch contains a fixed number of events, independent of real-world time.

**Definition 3.4** (DTDG: Time-based iteration). A DTDG is often expressed as a sequence of static graph snapshots sampled at regularly-spaced time intervals, i.e. as $\{\mathbf{G}_0, \mathbf{G}_1, ..., \}$, where $\mathbf{G}_i = \{\mathbf{V}_i, \mathbf{E}_i\}$ is a static graph at snapshot $i$ (Huang et al., 2024). In TGM, we achieve this by iterating

Table 2: Examples of common temporal graph operations represented as hooks, and their attributes.

| Hook Type | Neighbor Sampling | | Evaluation | Device Ops. | Analytics |
| | Recency | Uniform | TGB Eval | GPU Transfer | DOS Estimate |
|---|---|---|---|---|---|
| $\mathcal{R}$ (Requires) | {negatives} | {negatives} | $\emptyset$ | $\emptyset$ | $\emptyset$ |
| $\mathcal{P}$ (Produces) | {neighbors} | {neighbors} | {negatives} | $\emptyset$ | {DOS} |

with a time granularity $\hat{\tau}$ that is coarser than the native graph granularity. Iterating by time produces batches $\mathcal{G}|_{[t_0,t_i]}, \mathcal{G}|_{[t_i,t_{i+1}]}, \cdots$ where $|t_i - t_0| = |t_{i+1} - t_i| = \hat{\tau}$.

**Discretizing Temporal Graphs.** For snapshot-based models, it is often useful to process the graph at a coarser granularity than the native $\tau$ (e.g., daily instead of second-wise). Discretization converts the underlying network to this coarser timeline by collapsing duplicate edges within each time interval:

**Definition 3.5** (Time Granularity Discretization.). Let $\mathcal{G}$ be a temporal graph with native time granularity $\tau$. For any $\hat{\tau} \geq \tau$, the discretization operator:

$$\psi_r : (\mathcal{G}, \tau) \mapsto (\hat{\mathcal{G}}, \hat{\tau}) \tag{1}$$

maps $\mathcal{G}$ to coarser granularity $\hat{\tau}$, groups events into equivalence classes induced by $\hat{\tau}$ and applies a reduction operator $r$ to each class. The resulting graph $\hat{\mathcal{G}}$ contains one representative event per class. Figure 2 illustrates these time operations in TGM.

## 3.2 TGM Hooks and Recipes

In TGM, we formalize TGL workflows as compositions of data transformations called hooks. Each hook specifies certain batch attributes as inputs and outputs thus forming dependency relations between hooks, i.e. one hook produces an attribute that is required by another. A set of hooks forming a valid composition of hook signatures are considered as a recipe.

**Definition 3.6** (Materialized Batch). Let $G|_{\mathcal{T}}$ be a temporal subgraph. We denote by $B|_{\mathcal{T},\mathcal{A}}$ the materialized batch associated with a set of properties $\mathcal{A}$. Intuitively, $\mathcal{A}$ captures the attributes that enrich the slice of data, typically tensors required by a model (e.g. neighborhood information in message-passing architectures).

**Definition 3.7** (Hook). A hook $\phi_{\mathcal{R},\mathcal{P}}$ is a transformation on a materialized batch:

$$\phi_{\mathcal{R},\mathcal{P}} : \mathcal{B}|_{\mathcal{T},\mathcal{A}} \mapsto \mathcal{B}|_{\mathcal{T},\mathcal{A}\cup\mathcal{P}} \tag{2}$$

which declares a contract based on the attributes required on the input $\mathcal{R} \subset \mathcal{A}$, and the attributes produced $\mathcal{P}$, so that the batch transformed by $\phi$ has attributes $\mathcal{A} \cup \mathcal{P}$. Table 2 illustrates several common temporal graph operations expressed as hooks using the notation introduced here.

The real power of hooks is unlocked by composing their transformations to express complete temporal graph workflows. The notion of a hook recipe formalizes this.

**Definition 3.8** (Hook Recipe). A set of hooks $\{\phi^1_{\mathcal{R}_1,\mathcal{P}_1}, ..., \phi^k_{\mathcal{R}_k,\mathcal{P}_k}\}$ induces an ordering given by their dependencies:

$$\phi^i \to \phi^j \iff \mathcal{P}_i \cap \mathcal{R}_j \neq \emptyset \tag{3}$$

We call this a hook recipe if this dependency graph is acyclic and every required is satisfied, i.e. $\forall j, \mathcal{R}_j \subset \bigcup_{i<j} \mathcal{P}_i$. Thus, any hook recipe admits a valid ordering by topological sort. With this framework, exploring new research is simpler as complex workflows can be expressed with minimal boilerplate. Figure 3 illustrates how ML and analytics pipelines are represented as recipes in TGM.

## 4 TGM Software Library

We now describe the software implementation that realizes the framework described in the previous section. Figure 4 presents the high-level system design: the data layer is an immutable time-sorted coordinate format (COO) storage with lightweight graph views for efficient slicing; the execution

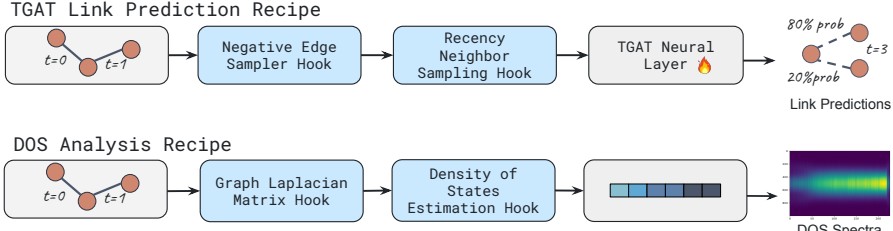

Figure 3: Example recipes in TGM: TGAT link prediction and Density of States Analysis. TGM provides a unified ecosystem supporting both representation learning and temporal graph analytics. The constituent hooks are modular, enabling reuse across different workflows within the community.

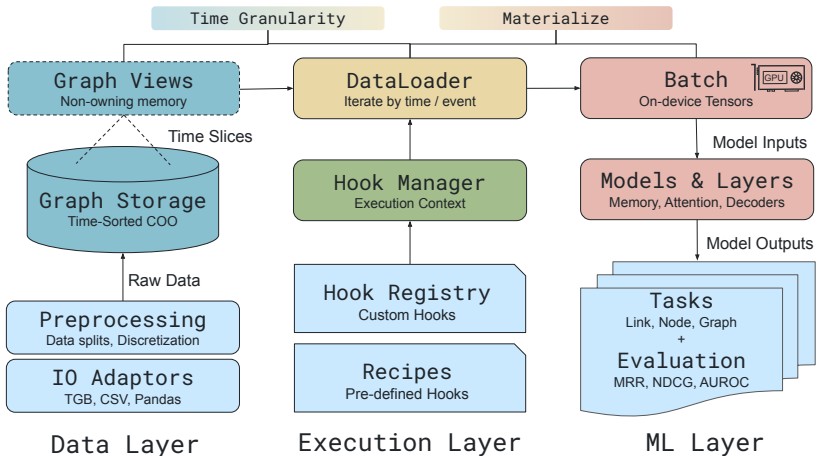

Figure 4: Three Layer Architecture of TGM: data layer (left), with IO adaptors and preprocessing, immutable COO graph storage, and lightweight sub-graph views; execution layer (middle), where users register custom hooks or apply pre-built recipes through the hook manager and dataloader to inject execution logic; and ML layer (right), where batches are materialized on device and used for node-, link-, or graph-level prediction. Light blue elements denote user-facing APIs.

layer is built around a hook manager that transparently performs complex transformations (e.g., temporal neighbors); and the ML layer materializes batches on-device for model computation. This separation of concerns yields workflows that are efficient and extensible, as we show in Section 5.3. Note that batch sizes significantly impact model performance (see Appendix G for further discussion).

**IO Adaptors and Data Preprocessing.** TGM streamlines experimentation by integrating the widely-used benchmark dataset: TGB (Huang et al., 2023a; Gastinger et al., 2024), in the form of IO Adapters, including loading, preprocessing, and train/validation/test splits. This allows researchers to start experiments immediately and compare models consistently with minimal overhead. Custom adapters are also supported via CSV and Pandas. Our design makes it straightforward to incorporate new benchmarks while ensuring consistent evaluation across all datasets (see Appendix D).

**Graph Storage and Graph Views.** The storage exposes an interface for graph queries, implemented using a time-sorted COO with a cached index. This enables binary search over timestamps, which is critical for recent-neighbor retrieval. The backend is designed for extension, allowing alternative layouts (Zhang et al., 2021; Sha et al., 2017) so future models can use the most efficient data structures for their workload. Backed by the storage, graph views provide lightweight, concurrency-safe access to temporal sub-graphs. Each view tracks time boundaries and encodes read-access through the time granularity abstraction. This enables TGM to perform both CTDG and DTDG-style loading, making it straightforward to study the effects of snapshot resolutions, as illustrated in Section 5.3. Our discretization is fully vectorized, enabling efficient snapshot creation, as demonstrated in Table 5.

**Hook Registry and Management.** Building on the graph abstractions, hooks are transformations that can be combined to create workflows (see Section s 3). Hooks process batches in chronological order to preserve temporality, while operations on events within the same batch, including neighbor sampling, device transfer, and negative edge generation, are executed in parallel, thus contributing to the efficiency of TGM. The HookManager handles shared state, resolves dependencies, and executes transparently during data loading. A key-value interface allows hooks to be registered under specific conditions (e.g., analytics hooks). We provide pre-defined recipes for common tasks such as TGB link prediction, helping new practitioners avoid common pitfalls like mismanaging state across splits.

**Diverse Model and Task Support.** TGM provides PyTorch modules tailored for TGL, including memory units, attention layers, and link decoders. With this, TGM implements a range of TG methods, from baselines like EdgeBank (Poursafaei et al., 2022a), to message passing-based models like TGAT (da Xu et al., 2020), and frontier models like DyGFormer (Yu et al., 2023a) and TPNet (Lu et al., 2024b). Crucially, learnable components are decoupled from graph management, making it easy for researchers to prototype new models.

```python
from tgm import DGData, DGraph, RecipeRegistry
from tgm.loader import DGDataLoader
from tgm.constants import RECIPE_TGB_LINK

# Load TGB Dataset and split data
train, ... = DGData.from_tgb("tgbl-wiki").split()

# Create storage-backed views over train split
train_dg = DGraph(train, device='cuda')

# Build TGB Link Property Prediction Recipe
manager = RecipeRegistry.build(RECIPE_TGB_LINK)
manager.register(...) # Register custom hooks
```

```python
# Inject hook manager into our data loader
loader = DGDataLoader(train_dg, manager, ...)

# Create model and optimizer
model, optimizer = ...

for epoch in range(NUM_EPOCHS):
  with manager.activate("train"):
    for batch in loader:
      loss = compute_loss(model(batch))
      loss.backward(); optimizer.step()

  manager.reset_state() # Reset hooks after epoch
```

Figure 5: Example workflow in TGM. Left: dataset loading, graph creation, and hook registration; Right: manager injection, model setup, and training loop with automatic hook activation. Highlighted code maps to system components from Figure 4.

**Streamlined TGL Workflows.** Figure 5 provides a high-level overview of a typical workflow in TGM, showing how data preparation, graph creation, hook registration, and training are orchestrated. Registered hooks dynamically inject behaviour during data loading, ensuring models automatically receive the appropriate tensors. This unifies the model interface and defines which batch attributes each model consumes. The manager reset method exposes a simple API for clearing the state of active hooks. Complex workflows can be implemented by registering hooks under key-value pairs.

**Robust and Research-Ready Infrastructure.** Finally, TGM is built following modern software engineering practices to ensure reliability, maintainability, and ease of use. We use type hinting throughout the codebase, which unifies model APIs and improve usability. Continuous integration pipelines run end-to-end tests on all layers, hooks, and graph APIs with test coverage to ensure correctness. Performance monitoring utilities can track GPU usage with support for tools such as FlameProf (Bobrov, 2017) to help identify bottlenecks. We also provide detailed tutorials, documentation, and examples for link, node and graph tasks. Overall, TGM provides a high-quality, research-ready platform that lowers the barrier to TG research while supporting efficient experimentation.

## 5 EXPERIMENTS

In this section, we evaluate TGM efficiency and research extensibility. Correctness results are reported in Section 5.2, where we show that TGM reproduces prior library performance. The appendix also includes peak memory measurements ( B.2) and a detailed runtime breakdown ( B.3) collected with TGM 's profiling tools. All experiments share the TGB (Huang et al., 2023a; Gastinger et al., 2024) 70/15/15 chronological data splits for train/validation/test, and hyperparameters as reported in Table 14. Runtimes are benchmarked under the same compute resources as described in Section F.

### 5.1 EFFICIENCY BENCHMARK

We evaluate TGM on two standard TGL tasks: dynamic link property prediction and dynamic node property prediction. Since graph discretization is a core operation in DTDG methods, we additionally

Table 3: Training time per epoch (seconds, ↓) for link property prediction. The **First** and *Second* best results are highlighted (× marks unsupported). TGM achieves competitive performance to the system-optimized TGLite library on TGAT and TGN models while supporting a broader range of architectures, and consistently outperforms the widely used research library DyGLib across all datasets and models, delivering a $4.4\times$ speedup on the transformer-based DyGFormer architecture.

| Model | tgbl-wiki | | | | tgbl-subreddit | | | | tgbl-lastfm | | | |
|---|---|---|---|---|---|---|---|---|---|---|---|---|
| | TGM | DyGLib | TGLite | TGL | TGM | DyGLib | TGLite | TGL | TGM | DyGLib | TGLite | TGL |
| TGAT | *6.97* | 41.24 | **4.85** | 10.00 | *28.23* | 182.21 | **25.00** | 53.25 | *55.32* | 349.31 | **38.00** | 85.12 |
| TGN | *10.59* | 63.37 | **6.80** | 23.32 | *61.25* | 287.06 | **60.50** | 125.23 | **91.23** | 392.98 | *92.93* | 250.00 |
| DyGFormer | **17.00** | *75.10* | × | × | **72.29** | *326.60* | × | × | **142.40** | *633.99* | × | × |
| TPNet | **12.28** | × | × | × | **49.79** | × | × | × | **97.23** | × | × | × |
| GCLSTM | **3.56** | × | × | × | **9.17** | × | × | × | **140.69** | × | × | × |
| GCN | **2.50** | × | × | × | **7.88** | × | × | × | **96.89** | × | × | × |

benchmark the efficiency of TGM in this setting. All datasets are stored in CPU host memory and transferred to GPU when required. Full experiment details, including model hyperparameters and compute resources, are provided in Appendix F.

**Link Property Prediction.** We benchmark TGM against state-of-the-art libraries on the dynamic link property prediction task using three standard datasets: `tgbl-wiki`, `tgbl-subreddit`, and `tgbl-lastfm`. Competing baselines include DyGLib (Yu et al., 2023a), TGL (Zhou et al., 2022b), and TGLite (Wang & Mendis, 2024), all of which are designed primarily for continuous-time models.

Table 3 reports training time per epoch across models implemented in TGM and competing libraries. First, TGM uniquely supports the widest range of architectures, spanning both CTDG and DTDG methods. In particular, DTDG models such as GCLSTM and GCN are supported via graph discretization and iterate-by-time functionality, and TGM is the only library with native support for TPNet (Lu et al., 2024a), the state-of-the-art link prediction model on TGB as of September 2025. Second, TGM consistently ranks among the top two fastest implementations across datasets and models. It outperforms DyGLib and TGL in all cases, and is only slightly behind the highly specialized TGLite on TGAT and TGN. For example, TGM achieves a $4.4\times$ speedup over the alternative DyGFormer implementation on `tgbl-wiki`. A key driver of performance is our fully vectorized recency sampler, implemented with a circular buffer in PyTorch-native code, which enables cache-friendly memory access. Finally, TGM offers native support for TGB evaluation, the standard benchmark protocol. Appendix B shows that TGM can be up to $246\times$ faster than DyGLib for TGN on `tgbl-wiki`, owing to batch-level de-duplication and efficient data handling: while DyGLib repeatedly samples neighbors for each prediction, TGM samples once per batch. By contrast, TGL and TGLite do not support this one-vs-many evaluation, limiting their benchmarking robustness compared to TGM.

**Node Property Prediction.** We benchmark TGM on the dynamic node property prediction task, comparing against both DyGLib and the native TGB implementations on the `tgbn-trade` and `tgbn-genre` datasets. TGL and TGLite do not support this task. Table 4 reports training time per epoch. Compared to DyGLib, TGM achieves up to a $10\times$ speedup for TGN on `tgbn-trade` while reducing training time by 80 seconds on `tgbn-genre`. Moreover, TGM is the only library supporting node property prediction for DTDG models: GCLSTM, GCN, and TGCN. Note, we encountered an `OOM` while running DyGLib on `tgbn-genre` with our 64GB RAM allocation (see Appendix F), requiring 256GB of memory to produce the results reported in Table 4.

Table 4: Training time per epoch (seconds, ↓) for dynamic node property prediction. The **First** and *Second* best results are highlighted (× marks unsupported). TGM has the best all-around performance and uniquely supports message-passing (TGN), snapshots-based (e.g. TGCN), and transformer-based (DyGFormer) models.

| Model | tgbn-trade | | | tgbn-genre | | |
|---|---|---|---|---|---|---|
| | TGM | DyGLib | TGB | TGM | DyGLib | TGB |
| TGN | *12.94* | 19.37 | **11.07** | **208.88** | 918.46 | *281.36* |
| DyGFormer | **16.24** | *117.13* | × | **70.89** | *3539.95* | × |
| P.F. | **0.41** | 2.09 | *0.78* | **38.15** | 41.73 | 35.58 |
| TGCN | **0.85** | × | × | **17.27** | × | × |
| GCLSTM | **0.88** | × | × | **17.71** | × | × |
| GCN | **0.80** | × | × | **17.21** | × | × |

Table 6: The choice of snapshot time granularity significantly affects link prediction performance. Reported metric is MRR (↑) with the **First** and *Second* best result for each dataset highlighted.

| Time Gran. | tgbl-wiki | | | tgbl-subreddit | | |
|---|---|---|---|---|---|---|
| | GCN | T-GCN | GCLSTM | GCN | T-GCN | GCLSTM |
| Hourly | $0.510 \pm 0.001$ | $0.509 \pm 0.004$ | $0.395 \pm 0.022$ | **$0.529 \pm 0.012$** | *$0.374 \pm 0.004$* | $0.219 \pm 0.003$ |
| Daily | **$0.702 \pm 0.007$** | *$0.540 \pm 0.008$* | $0.372 \pm 0.017$ | $0.266 \pm 0.007$ | $0.231 \pm 0.003$ | $0.212 \pm 0.004$ |
| Weekly | $0.393 \pm 0.005$ | $0.330 \pm 0.009$ | $0.323 \pm 0.010$ | $0.191 \pm 0.002$ | $0.212 \pm 0.001$ | $0.206 \pm 0.004$ |

**Graph Discretization.** Enabling DTDG models on CTDG tasks requires discretizing the original graph into snapshots. We compare TGM's implementation with that of UTG (Huang et al., 2024). Table 5 shows that TGM achieves dramatic speedups, up to $433\times$ on LastFM. This improvement stems from a fully vectorized, PyTorch-native implementation that avoids cache-unfriendly Python dictionaries and other overheads common in prior repositories. This result underscores our commitment to high-performance, research-ready tooling, setting TGM apart from existing libraries in efficiency and engineering standards.

Table 5: Discretization Latency to Hourly Snapshots (seconds, ↓). TGM has substantial speedups due to our vectorized, PyTorch-native implementation.

| Dataset | UTG | TGM | Speedup |
|---|---|---|---|
| tgbl-wiki | 1.94 | 0.04 | 49.62× |
| tgbl-subreddit | 8.83 | 0.21 | 41.63× |
| tgbl-lastfm | 19.94 | 0.05 | 433.39× |

## 5.2 CORRECTNESS TESTS

Table 7 reports MRR performance on texttttgbl-wiki for dynamic link property prediction, as well as NDCG on tgbn-trade for node property prediction. We cross-reference these results with TGB-reported performance and find that all models fall within the expected range. Note that we did not perform hyperparameter optimization but instead used the parameters listed in Table 14. On tgbl-wiki, CTDG models outperform DTDG baselines, with TPNet achieving the highest validation and test MRR, followed by DyGFormer and TGN. In contrast, for node property prediction, DTDG models, particularly GCLSTM and GCN, achieve the best held-out NDCG. These results highlight a complementary strength: CTDG models excel in link prediction, while DTDG models are more effective for node-level prediction.

Table 7: Performance on tgbl-wiki and tgbn-trade datasets. Numbers are mean ± std over 3 runs. The **First** and *Second* best results are highlighted (– marks unsupported)

| Category | Model | tgbl-wiki | | tgbn-trade | |
|---|---|---|---|---|---|
| | | Validation MRR (↑) | Test MRR (↑) | Validation NDCG (↑) | Test NDCG (↑) |
| **Baselines** | Edgebank | 0.495 | 0.527 | — | — |
| | P.F. | — | — | 0.860 | 0.855 |
| **DTDG** | GCN | **$0.465 \pm 0.013$** | **$0.410 \pm 0.019$** | *$0.670 \pm 0.013$* | *$0.629 \pm 0.009$* |
| | GCLSTM | *$0.402 \pm 0.016$* | $0.364 \pm 0.015$ | **$0.761 \pm 0.003$** | **$0.692 \pm 0.002$** |
| | TGCN | $0.400 \pm 0.017$ | $0.332 \pm 0.004$ | $0.515 \pm 0.006$ | $0.458 \pm 0.007$ |
| **CTDG** | TGAT | $0.380 \pm 0.013$ | $0.322 \pm 0.013$ | $0.380 \pm 0.006$ | $0.309 \pm 0.002$ |
| | TGN | $0.660 \pm 0.008$ | $0.527 \pm 0.008$ | *$0.393 \pm 0.001$* | **$0.329 \pm 0.002$** |
| | GraphMixer | $0.610 \pm 0.010$ | $0.567 \pm 0.018$ | — | — |
| | DyGFormer | *$0.743 \pm 0.006$* | *$0.712 \pm 0.009$* | $0.386 \pm 0.0012$ | *$0.312 \pm 0.0003$* |
| | TPNet | **$0.771 \pm 0.033$** | **$0.747 \pm 0.037$** | **$0.398 \pm 0.0034$** | $0.289 \pm 0.0030$ |

## 5.3 TGM RESEARCH EXPERIMENTS

In addition to its efficiency, TGM is designed as a flexible framework for exploring research questions in temporal graph learning. By supporting both CTDG and DTDG methods, along with native time conversions and composable hooks, TGM allows researchers to implement and test novel ideas effortlessly. We ran all three example experiments using a single script, which we include in our anonymized code release. These experiments investigate the following questions: RQ1:

Table 8: Binary classification task predicting whether the next daily snapshot will see an increased number of edges. Reported metric is AUC (↑) with the **First** and *Second* best results highlighted.

| | TGB | | MiNT | | |
|---|---|---|---|---|---|
| Model | tgbl-wiki | tgbl-subreddit | ADX | ARC | MIR |
| P.F. | $0.018 \pm 0.058$ | **$0.617 \pm 0.047$** | $0.397 \pm 0.010$ | *$0.625 \pm 0.022$* | $0.365 \pm 0.011$ |
| T-GCN | **$0.667 \pm 0.083$** | *$0.600 \pm 0.147$* | **$0.897 \pm 0.019$** | **$0.901 \pm 0.020$** | **$0.900 \pm 0.008$** |
| GCLSTM | $0.567 \pm 0.047$ | $0.526 \pm 0.020$ | *$0.588 \pm 0.047$* | $0.466 \pm 0.019$ | *$0.656 \pm 0.046$* |
| GCN | *$0.577 \pm 0.053$* | $0.200 \pm 0.000$ | $0.503 \pm 0.049$ | $0.608 \pm 0.061$ | $0.555 \pm 0.033$ |

How accurately can we predict the future evolution of a graph property? RQ2: How does the time granularity of graph snapshots impact DTDG performance on a continuous-time graph? RQ3: How do batching strategies, by fixed edges versus by time, affect the performance of a CTDG model?

**RQ1: Future evolution of a graph property.** Graph-level tasks require grouping edges into snapshots. The ability to natively support iteration by time is unique to TGM, allowing researchers to explore research questions in dynamic graph property prediction. As shown in Table 8, we benchmark models on predicting whether future snapshots on tgbl-wiki, tgbl-subreddit and transaction networks from the MiNT benchmark ( ADX, ARC and MIR), will grow or shrink, a critical problem for network evolution (Shamsi et al., 2025). The results highlight the sensitivity of model performance to temporal granularity: T-GCN performs best on tgbl-wiki with an AUC of 0.667, while P.F surprisingly excel on tgbl-subreddit with an AUC of 0.617. On MiNT, T-GCN consistently achieves the best scores with average AUC of 0.899 across three token networks.

**RQ2: Effect of Time Granularity for DTDG methods.** Table 6 demonstrates that the choice of snapshot granularity, i.e. hourly, daily, or weekly, has a substantial impact on the performance of snapshot-based temporal graph models. On the Wikipedia dataset, the impact is particularly pronounced: GCN's MRR increases by 30% when moving from weekly to daily snapshots, while T-GCN and GCLSTM improve by 21% and 5%, respectively. On Reddit, the same trend is observed, though less extreme: GCN achieves 0.529 MRR with hourly snapshots, dropping to 0.191 with weekly snapshots. These results underscore the importance of selecting an appropriate snapshot granularity for DTDG models. TGM makes this process effortless, allowing users to adjust the time granularity with a single line of code, treating it effectively as a hyperparameter.

**RQ3: Effect of Batch Size for CTDG methods.** Our analysis reveals that the configuration of the evaluation process itself is a critical, yet previously overlooked, hyperparameter in temporal graph learning. As demonstrated in Table 9, the choice of validation batch size and temporal unit significantly impacts the reported performance of the TGAT model on link prediction. Note that when iterating by time, the number of edges in each batch is different, however, each batch spans a fixed amount of time instead. We observe a pronounced degradation in MRR with larger batch sizes and coarser temporal units (e.g., Day versus Hour). TGM supports flexible temporal batching via our graph formulation, enabling the investigation of batch size at test time.

Table 9: The choice of validation batch size and batch unit affects the performance of TGAT link prediction on tgbl-wiki. **First** and *Second* best are highlighted.

| | Size/Unit | Test MRR (↑) |
|---|---|---|
| Batch size | 1 | **$0.449 \pm 0.001$** |
| | 50 | $0.414 \pm 0.006$ |
| | 100 | *$0.414 \pm 0.004$* |
| | 200 | $0.403 \pm 0.004$ |
| Batch unit | Hour | $0.402 \pm 0.012$ |
| | Day | $0.349 \pm 0.004$ |

## 6 CONCLUSION

We present TGM, a modular and efficient framework for temporal graph learning built around a novel hook formalism. By decoupling graph operations from model logic, TGM enables rapid prototyping and code reuse, unifying CTDG and DTDG methods under a single research-ready library. Efficiency-wise, TGM is highly competitive and on average $7.8\times$ faster in training than the widely used DyGLib. We ultimately envision TGM as a foundation for a shared ecosystem where models, hooks, and analytics can be seamlessly composed and reused, accelerating TGL research.

## ACKNOWLEDGEMENTS

This research was supported by the Canadian Institute for Advanced Research (CIFAR AI chair program), the EPSRC Turing AI World-Leading Research Fellowship No. EP/X040062/1 and EPSRC AI Hub No. EP/Y028872/1. Shenyang Huang was supported by the Natural Sciences and Engineering Research Council of Canada (NSERC) Postgraduate Scholarship Doctoral (PGS D) Award and Fonds de recherche du Québec - Nature et Technologies (FRQNT) Doctoral Award.

This research was also enabled in part by compute resources provided by Mila (mila.quebec).

## REPRODUCIBILITY STATEMENT

We provide a complete codebase to enable full reproducibility at `https://github.com/tgm-team/tgm`. All experiments use fixed random seeds, and full hyperparameters are listed in Table 14. The Python environment was built deterministically and managed with the `uv` package manager. Benchmarks were conducted in isolated SLURM jobs, and additional details on compute resources and experimental setup is provided in Appendix F.

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

Table 10: Validation time per epoch (seconds, ↓) for link property prediction (top) and node property prediction (bottom). The **First** and *Second* best results are highlighted (× marks unsupported). `OOT` indicates that a single validation epoch did not complete after 3 hours.

| Model | Wikipedia | | | | Reddit | | | | LastFM | | | |
|---|---|---|---|---|---|---|---|---|---|---|---|---|
| | TGM | DyGLib | TGLite | TGL | TGM | DyGLib | TGLite | TGL | TGM | DyGLib | TGLite | TGL |
| EdgeBank | **11.08** | *950.05* | × | × | **50.01** | *134.55* | × | × | **223.01** | *470.08* | × | × |
| TGAT | **532.89** | *2898.53* | × | × | **2241.70** | OOT | × | × | **4163.20** | OOT | × | × |
| TGN | **13.84** | *3404.82* | × | × | **60.30** | OOT | × | × | **112.23** | OOT | × | × |
| DyGFormer | **6.97** | *6125.05* | × | × | **1856.78** | OOT | × | × | **3554.252** | OOT | × | × |
| TPNet | **408.91** | × | × | × | **1735.71** | × | × | × | **3308.91** | × | × | × |
| GCLSTM | **11.92** | × | × | × | **51.68** | × | × | × | **110.16** | × | × | × |
| GCN | **11.70** | × | × | × | **50.88** | × | × | × | **102.56** | × | × | × |

| Model | Trade | | | Genre | | |
|---|---|---|---|---|---|---|
| | TGM | DyGLib | TGB | TGM | DyGLib | TGB |
| P.F. | **0.06** | 1.35 | *0.15* | **6.02** | 8.56 | *6.66* |
| TGN | *2.44* | 2.54 | **2.19** | **25.37** | 106.34 | *58.13* |
| DyGFormer | **3.49** | *21.13* | × | **11.78** | *588.69* | × |
| TGCN | **0.08** | × | × | **6.39** | × | × |
| GCLSTM | **0.07** | × | × | **6.48** | × | × |
| GCN | **0.07** | × | × | **6.46** | × | × |

# A LLM Usage

We acknowledge the use of LLMs to assist in polishing the writing of this paper. All content, ideas, and results are our own. The LLM helped improve clarity, grammar, style, and LaTeX formatting.

# B Additional Results

## B.1 Validation Latency Benchmarks

In Table 10, we report the TGB validation evaluation time per epoch for TGM and other libraries. Note that TGM supports highly optimized evaluation time for the robust TGB link prediction evaluation when compared to DyGLib. TGM consistently outperforms the widely used research library DyGLib across datasets and models. TGLite and TGL do not support the one-vs-many TGB-based link prediction evaluation (Gastinger et al., 2024).

## B.2 Peak GPU Usage

Table 11 shows the peak GPU memory usage of each model across three standard datasets. Lightweight models such as GCN and GCLSTM consume minimal memory, making them efficient choices for resource-constrained environments, whereas larger architectures like GraphMixer and DyGFormer require significantly more GPU memory. This comparison highlights the trade-offs between model size and memory efficiency, providing a practical reference for selecting models in temporal graph learning tasks.

Table 11: Peak GPU memory usage (GB) per model on different datasets.

| Model | tgbl-wiki | tgbl-subreddit | tgbl-lastfm |
|---|---|---|---|
| TGAT | 0.55 | 0.57 | 0.30 |
| TGN | 0.67 | 0.81 | 0.11 |
| GraphMixer | 2.61 | 2.62 | 2.62 |
| DyGFormer | 1.34 | 1.36 | 1.03 |
| TPNet | 1.37 | 1.47 | 1.15 |
| GCLSTM | 0.01 | 0.18 | 0.07 |
| GCN | 0.01 | 0.09 | 0.05 |

## B.3 CProfiler Model Breakdown

Table 12 shows a runtime decomposition of TGAT on the `LastFM` dataset. The largest costs arise from the backward pass (25.8%), model forward (26.5%), and optimizer updates (19.1%), together accounting for over 70% of total runtime. Within data loading (26.5%), hook execution (15.1%) and graph materialization (11.4%) dominate, with the recency sampler alone contributing 13.2%. Inside TGAT forward, attention layers (14.7%) and MLPs (6.0%) form the bulk of computation, while time

encoding adds 3.5%. Using a profiler in this way helps researchers and practitioners identify which components are the main bottlenecks and prioritize optimizations accordingly.

Table 12: Breakdown of TGAT runtime on `LastFM` dataset.

| Category | Component | Percent (%) |
|---|---|---|
| Data Loading | Hook execution | 15.09 |
| | \|-- Recency sampler | 13.19 |
| | \| \|-- Get neighbors | 7.76 |
| | \| \|-- Update circular buffer | 5.43 |
| | \|-- Other hooks | 1.90 |
| | Graph materialization | 11.40 |
| Model Forward | TGAT forward | 24.20 |
| | \|-- Attention layers | 14.70 |
| | \|-- MLP layers | 5.96 |
| | \|-- Time encoding | 3.54 |
| | Other forward (decoders) | 2.30 |
| Optimization | Backward pass | 25.80 |
| | Optimizer (Adam) | 19.10 |
| | Loss computation | 0.62 |
| Other | - | 1.61 |

## C  ADDITIONAL BACKGROUND: DTDG VS. CTDG

As defined in Section 3, a temporal graph is a graph whose structure and attributes evolve over time, capturing not only the relationships between entities but also the dynamics of their interactions. Unlike static graphs, which provide a single snapshot of connectivity, temporal graphs represent edges (and sometimes nodes) as time-stamped events or intervals, enabling modelling of when and how relationships form, change, or disappear. Temporal graph neural networks are typically categorized into two types: continuous-time dynamic graph (CTDG) methods and discrete-time dynamic graph (DTDG) methods. Section C.1 and Section C.2 provide further information about common approaches from each category.

### C.1  DTDG METHODS

DTDG, or snapshot-based methods, take as input a sequence of graph snapshots, each representing the state of the temporal graph at discrete time intervals (e.g., hours or days). These approaches process each snapshot as a whole, typically using a graph learning model, and employ mechanisms to capture temporal dependencies across snapshots.

The majority of DTDG methods consist of two main components: a spatial encoder, commonly GNN-based, and a temporal encoder, usually an RNN or one of its variants. Given a snapshot $G_i$, a spatial representation is learned, $Z_i = f(V_i, E_i)$, where $f$ is a trainable or non-trainable function that takes the graph structure of the current snapshot and returns either node-level representations in $G_i$ or a representation of the entire snapshot. GCN (Kipf & Welling, 2017) is used as $f$ in TGCN (Zhao et al., 2019), EvolveGCN (Pareja et al., 2020), and GCLSTM (Chen et al., 2018). In contrast, GraphPulse (Shamsi et al., 2024) encodes a whole-graph representation by extracting topological features from both the original graph $G_i$ and a transformed version $G_i'$, using Topological Data Analysis (TDA). The concatenation of the features from $G_i$ and $G_i'$ serves as the graph-level representation for downstream property prediction tasks.

To capture temporal dependencies across snapshots, an RNN or one of its variants (e.g., GRU or LSTM) is typically employed. These are applied either to the sequence of snapshot representations $Z_i$ (Zhao et al., 2019; Chen et al., 2018; Shamsi et al., 2024) or directly to the evolving parameters of the GCN (Pareja et al., 2020).

## C.2 CTDG METHODS

In contrast, CTDG methods operate on a continuous stream of edges and can make predictions at arbitrary timestamps. They update internal representations incrementally as new interactions arrive, incorporating fresh information into predictions. For computational efficiency, the edge stream is usually partitioned into fixed-size batches, with predictions performed sequentially per batch; once predictions are made, the corresponding edges are revealed to the model. Unlike DTDG methods, CTDG approaches do not rely on snapshots; instead, they maintain evolving node representations and sample temporal neighborhoods around nodes of interest for prediction.

**Temporal Message Passing.** The temporal message passing framework is a neighbourhood aggregation scheme which recursively computes a latent representation by forwarding messages to temporal neighbours. Formally, if $\mathcal{N}^k(s)$ denotes the k-hop neighbourhood of node $s$ in the dynamic graph $\mathcal{G}$, then the *temporal neighbourhood* $\mathcal{N}_t^k(s)$ is given by restricting neighbours to edge events chronologically before time $t$:

$$\mathcal{N}_t^k(s) = \{(s, d, t') \in \mathcal{N}^k(s) : t' \leq t\} \tag{4}$$

The combination of temporal and topological constraints makes efficient neighbourhood particularly challenging, requiring complex hierarchical data structures and cache-aware programming to sustain high-throughput on GPU stream multiprocessors Zhang et al. (2021); Sha et al. (2017). We bypass the insertion and deletion complexity by assuming the entire graph structure is read-only. Temporal message proceeds by creating and passing messages between such sub-neighorhoods. In particular, messages are created by concatenating embeddings, aggregating embeddings across temporal neighbourhoods, then updating the new hidden representation. Such information flow occurs concurrently for each event in a batch of data.

**Time-Encoding and Memory-Based Learning.** Time-encoding models use a shift-invariant model $\psi : T \rightarrow \mathbb{R}^{d_t}$ that maps a real-valued time stamp into a $d_t$-dimensional vector (e.g. TGAT da Xu et al. (2020) use time-encoders like Time2Vec Kazemi et al. (2019)). This encoding is then passed through modified self-attention blocks or feedforward layers. *Memory-based* models, such as TGN Rossi et al. (2020), utilize a fixed-bandwidth memory module that compresses relevant information for each node and updates it over time. EdgeBank Poursafaei et al. (2022a) is a non-parametric, memory-based method that memorizes and predicts new links at test time based on their occurrence in the training data.

## D DATASET DETAILS

In this work, we conduct experiments on Wikipedia (obtained from the TGB Huang et al. (2023a), where the dataset can be downloaded along with the package from TGB website), Reddit, LastFM, datasets, obtained from Poursafaei et al. (2022b); these can be downloaded from `https://zenodo.org/records/7213796#.Y8QicOzMJB2`. These datasets span a variety of real-world domains, providing a broad testbed for evaluating temporal graph models. Detailed information about these datasets are as follows.

- **Wikipedia** is a bipartite interaction network that captures editing activity on Wikipedia over one month. The nodes represent Wikipedia pages and their editors, and the edges indicate timestamped edits. Each edge is associated with a 172-dimensional LIWC feature vector derived from the text.
- **Reddit** models user-subreddit posting behaviour over one month. Nodes are users and subreddits, and edges represent posting requests made by users to subreddits, each associated with a timestamp. Each edge is associated with a 172-dimensional LIWC feature vector based on post contents.
- **LastFM** is a bipartite user–item interaction graph where nodes represent users and songs. Edges indicate that a user listened to a particular song at a given time. The dataset includes 1000 users and the 1000 most-listened songs over a one-month period. This dataset is not attributed.
- **Trade** represents the international agriculture trading network between UN nations from 1986 to 2016. Nodes are countries and edges capture the annual sum of agriculture trade values from one country to another. The task is to predict each nation's trade proportions in the following year.
- **Genre** is a bipartite, weighted network connecting users to music genres based on listening history. Edges indicate the proportion of a song belonging to a genre that a user listens to, aggregated weekly. The task is to predict user-genre interactions in the next week, capturing evolving user preferences for music recommendation.

Table 13: Dataset statistics.

| Dataset | # Nodes | # Edges | # Unique Edges | # Unique Steps | Surprise | Duration |
|---------|---------|---------|----------------|----------------|----------|----------|
| Wikipedia | 9,227 | 157,474 | 18,257 | 152,757 | 0.108 | 1 month |
| Reddit | 10,984 | 672,447 | 78,516 | 669,065 | 0.069 | 1 month |
| LastFM | 1,980 | 1,293,103 | 154,993 | 1,283,614 | 0.35 | 1 month |
| Trade | 255 | 468,245 | 468,245 | 32 | 0.023 | 30 years |
| Genre | 1,505 | 17,858,395 | 17,858,395 | 133,758 | 0.005 | 1 month |

## E  TEMPORAL GRAPH MODELS SUPPORTED IN TGM

TGM is a research-driven library providing implementations of state-of-the-art temporal graph learning models. At the time of writing, TGM includes the following models:

**Persistent Forecast.** A simple baseline that predicts the future state of each node or edge by assuming it remains unchanged from the most recent observation. Despite its simplicity, it often serves as a strong baseline for dynamic node property prediction.

**EdgeBank.** Poursafaei et al. (2022a) Maintains a memory bank of historical edges and uses them to make predictions. By storing and sampling past interactions, EdgeBank leverages temporal patterns without explicit node embedding updates, providing a lightweight but effective approach for dynamic link prediction.

**TGAT.** da Xu et al. (2020) proposed to model dynamics node representations with TGAT layer, which is a combination of the graph attention mechanism with a time encoding function based on Bochner's theorem, which provides a continuous functional mapping from time to a vector space. This allows TGAT to efficiently learn from temporal neighbourhood features with the aid of a self-attention mechanism and temporal dependencies encoded by the time encoding function.

**TGN.** Rossi et al. (2020) proposed an event-based model that is a combination of a memory module, message aggregator, message updater and embedding module. In particular, the memory module maintains evolving memory for each node and updates this memory when the node is observed to be involved in an interaction, which is achieved by a message function, message aggregator, and message updater. Finally, the embedding model is used to compute the representation of nodes.

**GCN.**(Kipf & Welling, 2017) Standard Graph Convolutional Network applied on static snapshots to encode structural information. Node features are aggregated from neighbors and combined with self-features to produce updated embeddings at each snapshot. When used in temporal settings, GCNs process sequences of snapshots independently or in combination with temporal modules.

**GCLSTM.** To learn over a sequence of graph snapshots, Chen et al. (2018) proposed an end-to-end model named Graph Convolutional Long Short-Term Memory (GCLSTM) for dynamic link prediction. The LSTM serves as the backbone to capture temporal dependencies across graph snapshots, while a GCN is applied to each snapshot to encode structural dependencies between nodes. Specifically, two GCNs are used to update the hidden state and the cell state of the LSTM, and a multilayer perceptron (MLP) decoder maps the features at the current time step back to the graph space. This design enables GCLSTM to effectively handle both link additions and deletions.

**T-GCN.** Zhao et al. (2019) integrates GCNs with gated recurrent units to learn node embeddings over sequences of graph snapshots, capturing temporal and structural information jointly.

**GraphMixer.** (Sarıgün, 2023) A graph adaptation of MLP-Mixer architectures. It alternates between node-wise and feature-wise mixing layers to capture structural correlations across nodes and temporal correlations across features. By stacking multiple mixer layers, GraphMixer can model complex dependencies in dynamic graphs while remaining simple and parameter-efficient.

**DyGFormer.** Yu et al. (2023a) proposed a Transformer-based architecture for modeling dynamic graphs. DyGFormer consists of two key components: the Neighbour Co-occurrence Encoder and a Transformer. The Neighbour Co-occurrence Encoder leverages the recent first-hop neighbours of the source and destination nodes of an edge to capture correlations and compute relative embeddings. To enhance representation learning, Yu et al. (2023a) further introduced a patching technique that splits the source and destination node features, edge features, time embeddings (computed following

TGAT (da Xu et al., 2020)), and relative embeddings into multiple patches. These patches are then fed into the Transformer to generate node representations with respect to an edge.

**TPNet.** TPNet is composed of two main modules: Node Representation Maintenance and Link Likelihood Computation. Lu et al. (2024b) unifies existing relative encoding methods by introducing temporal walk matrices with an integrated time-decay function. These matrices establish a principled connection between relative encodings and temporal walks, offering a clearer framework for analyzing and designing temporal encodings. The time-decay effect further allows joint modelling of temporal and structural information. Since computing temporal walk matrices directly is computationally and memory intensive, TPNet employs a theoretically grounded random feature propagation mechanism to implicitly approximate and maintain them efficiently.

The TGM team is actively expanding the library to incorporate additional cutting-edge models, including TNCN (Zhang et al.), DyGMamba (Ding et al., 2024), NAT (Luo & Li, 2022), and TGNv2 (Tjandra et al.).

# F  COMPUTE RESOURCES AND EXPERIMENT DETAILS

**Compute**: Experiments were run on Ubuntu 20.04 with 64 GB RAM, 4 isolated AMD EPYC 7502 CPU cores, and a single 80 GB A100 GPU. Jobs were managed with SLURM to ensure isolated environments and no concurrent interference.

**Experiment Details**: We use the default TGB splits (Huang et al., 2023a; Gastinger et al., 2024), with hyperparameters listed in Table 14. For efficiency benchmarks, TGAT and TGN adopt the TGLite configuration (Wang & Mendis, 2024) for fairness. Other libraries were modified only minimally to measure latency, and TGLite/TGL times are taken directly from Fig. 6 of (Wang & Mendis, 2024). All DTDG methods discretized the `Trade` dataset to yearly snapshots, and the `Genre` dataset to weekly snapshots.

Table 14: Hyperparameters used for each model

| Parameter | Edgebank | TGAT | TGN | GCN | GCLSTM | TGCN | GraphMixer | DyGFormer | TPNet |
|---|---|---|---|---|---|---|---|---|---|
| Batch Size | 200 | 200 | 200 | 200 | 200 | – | 200 | 200 | 200 |
| Epochs | – | 10 | 30 | 30 | 30 | – | 10 | 5 | 10 |
| Learning Rate | – | 1e-4 | 1e-4 | 1e-3 | 1e-3 | 1e-3 | 2e-4 | 1e-4 | 1e-4 |
| Dropout | – | 0.1 | 0.1 | 0.1 | – | 0.1 | 0.1 | 0.1 | 0.1 |
| # Heads | – | 2 | 2 | – | – | – | – | 2 | – |
| # Neighbors | – | 20 | 10 | – | – | – | 20 | 32 | 32 |
| # Layers | – | 2 | 2 | 2 | 2 | 2 | 2 | 2 | 2 |
| Embedding Dim. | – | 100 | 100 | 128 | 256 | 128 | 128 | 172 | 172 |
| Time Dim. | – | 100 | 100 | – | – | – | 100 | 100 | 100 |
| Memory Dim. | – | – | 100 | – | – | – | – | – | – |
| Node Dim. | – | – | – | 256 | 256 | 256 | 100 | 128 | 128 |
| Sampling | – | Recency | Recency | – | – | – | Recency | Recency | Recency |
| Memory Mode | Unlimited | – | – | – | – | – | – | – | – |
| Time Gap | – | – | – | – | – | – | 2000 | – | – |
| Token Dim. Factor | – | – | – | – | – | – | 0.5 | – | – |
| Channel Dim. Factor | – | – | – | – | – | – | 4.0 | – | – |
| Channel Dim. | – | – | – | – | – | – | – | 50 | – |
| Patch Size | – | – | – | – | – | – | – | 1 | – |
| # Channels | – | – | – | – | – | – | – | 4 | – |
| # RP Layers | – | – | – | – | – | – | – | – | 2 |
| RP Time Decay | – | – | – | – | – | – | – | – | 1e-6 |
| RP Dim | – | – | – | – | – | – | – | – | $\log(|2 * E|)$ |

# G  EFFECT OF BATCH SIZE

TGL models often update their representation after a batch has been processed to incorporate the most recent information. Events within a batch are processed in parallel in TGM thus larger batch size leads to more efficient inference. Therefore, the batch size parameter becomes a trade-off between efficiency and model update frequency (or a model's staleness). In this work, we follow the TGB Huang et al. (2023a) evaluation procedure of using 200 edges per batch where applicable. With the time slicing operation in TGM, it is now possible to explore arbitrary or dynamic batch size where a batch is formed by slicing a graph in a custom time interval. An alternative way is to iterate by time

in TGM, which provides variable batch size in terms of number of edges and more empirical results are reported in Section 5.3. We believe that exploring the implication of batching is an important future direction and leave it as an interesting future work which can be built-upon in TGM.

## H  TGM TUTORIALS AND DOCUMENTATION

Ensuring reproducibility and ease of use is a top priority in TGM. We therefore provide a full documentation suite, including beginner-friendly tutorials and a comprehensive API specification. Below, we include excerpts from our tutorials.

# Temporal Graph Data in TGM

This tutorial shows the **core graph API** in TGM. By the end, you should understand how to:

- Construct and preprocess graph data (`DGData`)
- Split and discretize temporal datasets (`SplitStrategy`)
- Work with immutable graph views (`DGraph`)
- Train with batches (`DGBatch`)

We also highlight some important errors, caching behaviour, and best practices.

---

## 1. The Core Objects

TGM's graph API revolves around four main objects:

| Object | Description | Mutable | Device Semantics | Typical Usage |
|--------|-------------|---------|------------------|---------------|
| `DGData` | Mutable bulk dataset storage (IO, splits, transforms) | Yes | No | Ingesting datasets from disk, TGB, preprocessing |
| `DGraph` | Immutable graph view backed by storage engine | No | Yes | Main user-facing graph object |
| `DGBatch` | Materialized batches of tensors from a temporal slice of data | Yes | Yes | What dataloaders yield, input to models |

latest

| Object | Description | Mutable | Device Semantics | Typical Usage |
|---|---|---|---|---|
| `DGStorage` | Internal backend for graph data (non-user-facing) | No | Yes | Powers graph querying, caching, slice ops |

> **Note**: Users typically only interact with the first 3. `DGStorage` is internal and abstracted away. It is in our stream of work to build out more efficient storage backends for various workloads in the future.

## 2. Starting with `DGData`

`DGData` is your *main* entry point for working with temporal graph datasets. It's a dataclass that holds bulk storage of events, timestamps, features, and metadata.

Because it's mutable, you can freely transform and prepare it before moving to the immutable graph representation ( `DGraph` ).

**Features of `DGData`**

- Holds raw edge data ( `edge_index` , `edge_timestamps` )
- Holds *static node features*, *dynamic node features*, and *edge_features* (on CPU)
- Provides IO constructors (CSV, Pandas, TGB, pyTorch)
- Supports *temporal splitting* and *discretization*
- Ensures data is sorted chronologically, valid node ids, valid tensor shapes, etc.

See below for a summary of the data class attributes of `DGData` :

```
@dataclass
class DGData:
    """Container for dynamic graph data to be ingested by `DGStorage`.

    Stores edge and node events, their timestamps, features, and optional split
strategy.
    Provides methods to split, discretize, and clone the data.

    Attributes:
        time_delta (TimeDeltaDG | str): Time granularity of the
        timestamps (Tensor): 1D tensor of all event timestamps [num_edge_events +
num_node_events].
```

```
        edge_event_idx (Tensor): Indices of edge events within `timestamps`.
        edge_index (Tensor): Edge connections [num_edge_events, 2].
        edge_feats (Tensor | None): Optional edge features [num_edge_events,
D_edge].
        node_event_idx (Tensor | None): Indices of node events within
`timestamps`.
        node_ids (Tensor | None): Node IDs corresponding to node events
[num_node_events].
        dynamic_node_feats (Tensor | None): Node features over time
[num_node_events, D_node_dynamic].
        static_node_feats (Tensor | None): Node features invariant over time
[num_nodes, D_node_static].

    Raises:
        InvalidNodeIDError: If an edge or node ID match `PADDED_NODE_ID`.
        ValueError: If any data attributes have non-well defined tensor shapes.
        EmptyGraphError: If attempting to initialize an empty graph.

    Notes:
        - Timestamps must be non-negative and sorted; DGData will sort
automatically if necessary.
        - Cloning creates a deep copy of tensors to prevent in-place
modifications.
        """
```

See `tgm.data.DGData` for full reference.

# 3. Constructing DGData

You can build datasets in multiple ways. Let's look at each.

## 3.1 From TGB

This is most likely all you need. The Temporal Graph Benchmark (TGB) provides a suite of temporal graph datasets with diverse scales and properties. We natively support direct construction from all the `tgbl-` and `tgbn-` in TGM.

> **Note**: Temporal knowledge graph (TKG) and temporal hypergraph (THG) are not yet supported in TGM.
>
> **Note**: To load a TGB dataset, you must have the `py-tgb` package in your python env.

```
from tgm import DGData                                        ⌥ latest  ▾

# Load the Wikipedia dataset from TGB
data = DGData.from_tgb('tgbl-wiki')
```

```
print(data.time_delta) # TimeDelta('s', value=1)
print(data.edge_index.shape) # torch.Size([157474, 2])
print(data.dynamic_node_feats) # None, no dynamic node features in tgbl-wiki
print(data.static_node_feats) # None, no static node features in tgbl-wiki
```

> **TIP**: You can `print(data)` to see which features and events exist within the dataset.

## 3.2 Custom Datasets

If you have our own dataset in TGM, you can create a `DGData` object either `from_csv`, `from_pandas`, or directly from tensors. A brief overview of each is given below, consult the API reference for more details.

**From CSV**

Please consult our documentation for full description of our API. The table below summarizes the main pieces of data expected during construction. Note that analogous attributes are expected in the other IO constructors (e.g. `from_pandas`, `from_raw`)

| Attribute | Description | Type | Required | Note |
|---|---|---|---|---|
| `edge_file_path` | Path to CSV file containing edge data | `str \| pathlib.Path` | Yes | `edge_df` if using `from_pandas` |
| `edge_src_col` | Column name in edge file for src nodes | `str` | Yes | Cannot have ids matching `tgm.constants.PADDED_NODE_ID` |
| `edge_dst_col` | Column name in edge file for dst nodes | `str` | Yes | Cannot have ids matching `tgm.constants.PADDED_NODE_ID` |
| `edge_time_col` | Column name in edge file for edge times | `str` | Yes | Time must be non-negative |
| `node_file_path` | Path to CSV file containing | `str \| pathlib.Path` | No | `node_df` is using `from_pandas` |

 latest ▾

| Attribute | Description | Type | Required | Note |
|---|---|---|---|---|
| | dynamic node data | `th` | | |
| `node_id_co` `l` | Column name in node file for node event node ids | `str` | No, unless `node_file_pat` `h` is specified | Cannot have ids matching `tgm.constants.` `PADDED_NODE_ID` |
| `node_time_c` `ol` | Column name in node file for node event node times | `str` | No, unless `node_file_pat` `h` is specified | Time must be non-negative |
| `dynamic_nod` `e_feats_co` `l` | Column name in node file for dynamic node features | `str` | No | |
| `static_node` `_feats_file` `_path` | Path to CSV file containing static node features | `str \|` `pathlib.Pa` `th` | No | `static_node_fe` `ats_df` if using `from_pandas` |
| `static_node` `_feats_col` | Column name in static node feats file for static node features | `str` | No, unless `static_node_f` `eats_file_pat` `h` is specified | |
| `time_delta` | Time granularity of the graph data | `TimeDeltaD` `G \| str` | Yes | Default to *event_ordered* granularity `'r'` |

A few key things to know:

- `time_delta`: defines how timestamps are interpreted on your custom

⎇ latest ▼

- The default is 'r' which entails event-ordered semantics. This means there is no real-world time unit assigned to your timestamps. This prevents from doing things like discretizing your data,

and iterating by temporal snapshots.

- More often than not, your timestamps have some semantics meaning (e.g. *seconds*, *days*, etc). In this case, you should specify the appropriate `time_delta` value. See our time management tutorial for more details.

- edge data:

- We expect an `edge_file_path` which is a csv file with `edge_src_col`, `edge_dst_col`, `edge_time_col` as a minimum.

- Your edge csv file may also contain `edge_feats_col` which are the edge features on your data

- dynamic node data (optional)

- If included, we expect a `node_file_path` which is a csv file with `node_id_col`, `node_time_col` as a minimum. These are your dynamic node events.

- Your dynamic node data csv file may also include `dynamic_node_feats_col`, which are the dynamic node features in your data.

- static node data (optional)

- If included, we expect a `static_node_feats_fil_path` which is a csv file with `static_node_feats_col`, the static node features for your dataset.

Internally, we perform various checks on the tensors shapes, node ranges, and timestamps values. If your data is well structured, everything should work. If you get an error message that is not intuitive, please let us know.

**From Pandas**

The API largely the same as above, except that we expected `edge_df`, `node_df`, and `static_node_feats_df` dataframes for the edge, dynamic node, and static node data respectively, instead of csv files.

```python
import pandas as pd

# Define Edge Data
edge_df = pd.DataFrame({
    'src': [2, 2, 1],
    'dst': [2, 4, 8],
    't': [1, 5, 10],
    'edge_feat': [torch.rand(5).tolist() for _ in range(3)], # Optional
})

# Define Dynamic Node Data (Optional)
dynamic_node_df = pd.DataFrame({
    'node': [2, 4, 6],
    't': [1, 2, 3],
    'dynamic_node_feat': [torch.rand(5).tolist() for _ in range(3)],
```

```
})

# Define Static Node Features (Optional)
static_node_df = pd.DataFrame({
    'static_node_feat': [torch.rand(11).tolist() for _ in range(9)]
})

dg = DGraph.from_pandas(
    edge_df=edge_df,
    edge_src_col='src',
    edge_dst_col='dst',
    edge_time_col='t',
    edge_feats_col='edge_feat',
    node_df=dynamic_node_df,
    node_id_col='node',
    node_time_col='t',
    dynamic_node_feats_col='dynamic_node_feat',
    static_node_feats_df=static_node_df,
    static_node_feats_col='static_node_feat',
    time_delta='s',  # second-wise granularity
)
```

**From Tensors**

If all your data is already in memory as `torch.Tensor` you can directly instantiate `DGdata` using
the class method `DGData.from_raw`:

```
import torch

# Define Edge Data
edge_index = torch.LongTensor([[2, 2], [2, 4], [1, 8]])
edge_timestamps = torch.LongTensor([1, 5, 20])
edge_feats = torch.rand(3, 5)  # optional edge features

# Define Dynamic Node Data (Optional)
node_timestamps = torch.LongTensor([1, 2, 3])
node_ids = torch.LongTensor([2, 4, 6])
dynamic_node_feats = torch.rand([3, 5])

# Define Static Node Features (Optional)
static_node_feats = torch.rand(9, 11)

data = DGData.from_raw(
    edge_timestamps=edge_timestamps,
    edge_index=edge_index,
    edge_feats=edge_feats,
    node_timestamps=node_timestamps,
    node_ids=node_ids,
    dynamic_node_feats=dynamic_node_feats,
    static_node_feats=static_node_feats,
```

```
    time_delta='s',  # second-wise granularity
)
```

## 3.3 Errors to know

- `tgm.exceptions.EmptyGraphError` : Raised when you try to construct a `DGData` object from empty data. This is probably not what you intended to do since downstream `DGraph` is immutable.
- `tgm.exceptions.InvalidNodeIDError` : Raised when you dataset contains `-1` as a node ID (reserved for padding).

# 4. Splitting `DGData`

After loading your data, you'll probably want to split your dataset into *train*, *validation*, and *test* splits. TGM provides a **strategy pattern** interface for different split strategies:

- `TemporalSplit` : Split by fixed timestamp boundaries
- `TemporalRatioSplit` : Split by ratio of both edge and node events
- `TGBSplit` : Pre-defined TGB data splits

> **Important**: The TGB data splits uses pre-defined event masks, to match the splits as per the TGB leaderboard. If you try to change this, you'll get a `ValueError` .

The split method is defined on `DGData` :

```
def split(self, strategy: SplitStrategy | None = None) -> Tuple[DGData, ...]:
    """Split the dataset according to a strategy.

    Args:
        strategy (SplitStrategy | None): Optional strategy to override the
            default. If None, uses `_split_strategy` or defaults to
`TemporalRatioSplit`.

    Returns:
        Tuple[DGData, ...]: Split datasets (train/val/test).

    Raises:
        ValueError: If attempting to override the split strategy for TGB datasets.

    Notes:
        - Splits preserve the underlying storage; only indices
    """
```

Splitting TGB Datasets

```
from tgm import DGData

# Load the Wikipedia dataset from TGB
data = DGData.from_tgb('tgbl-wiki')

# Split using native TGB masks
train_data, val_data, test_data = data.split()

# If you tried to override the split strategy, you'll get an error

from tgm.split import TemporalRatioSplit
split_strategy = TemporalRatioSplit(train=0.8, val=0.1, test=0.1)
_ data.split(strategy=split_strategy) # Raises ValueError
```

## 5. Discretizing `DGData`

In TGM, we do not enforce strict definition of continuous time (resp. discrete time) dynamic graph CTDG (resp. DTDG). Instead, as you have seen, we define graphs based on their time granularity. Therefore, the user is able to convert between event-based and snapshot based views of the underlying data. You can learn more about this in the UTG paper.

In TGM, we provide a method on `DGData` called `discretize` which allows you to coarsen your graph into different time granularities. The API looks like:

```
def discretize(
    self, time_delta: TimeDeltaDG | str | None, reduce_op: str = 'first'
) -> DGData:
    """Return a copy of the dataset discretized to a coarser time granularity.

    Args:
        time_delta (TimeDeltaDG | str | None): Target time granularity.
        reduce_op (str): Aggregation method for multiple events per bucket.
Default 'first'.

    Returns:
        DGData: New dataset with discretized timestamps and features.

    Raises:
        EventOrderedConversionError: If discretization is incompatible with event-
ordered granularity
        InvalidDiscretizationError: If the target granularity is finer than the
current granularity.
    """
```

⅄ latest ▾

> **Note**: This is only well defined if the DGData time delta is *time-ordered*. If you try discretizing an event-ordered dataset, you will get a `tgm.exceptions.EventOrderedConversionError`.
>
> **Note**: Discretization goes from finer time units (e.g. seconds) to coarse time units (e.g. hours). If your attempt to discretize in the other direction, you'll get a `tgm.exceptions.InvalidDiscretizationError`.

See our time management tutorial for more details on discretization and how it relates to `TimeDeltaDG`.

## 6. From `DGData` to `DGraph`

Once your dataset is ready to go, you can cast it to `DGraph`:

```python
from tgm import DGraph, DGData

data = DGData.from_tgb(...)
dg = DGraph(data, device=...)
```

Some things to note:

- `DGraph` is an immutable view over a temporal window of graph data.
- It is backed by `DGStorage` (internal engine). When you first create a `DGraph` as we did above, a new storage is created, and the view encapsulates the entire dataset.
- `DGraph` supports device semantics, you can choose what device your graph is on.

### DGraph Properties

Let's use our toy `DGData` we had above, cast to `DGraph` and inspect some of the properties of the entire dataset.

```python
data = DGData.from_raw(...) # As we had above
dg = DGraph(data) # Default to CPU

print(f'Start time              : {dg.start_time}') # 1
print(f'End time                : {dg.end_time}') # 10
print(f'Number of nodes         : {dg.num_nodes}') # 9
print(f'Number of edge events   : {dg.num_edges}') # 3
print(f'Number of timestamps    : {dg.num_timestamps}') # or len(dg): 5
print(f'Total events (edge+node) : {dg.num_events}') # 6
print(f'Edge feature dimension  : {dg.edge_feats_dim}') # 5
print(f'Static node feature dim  : {dg.static_node_feats_dim}') # 11
print(f'Dynamic node feature dim : {dg.dynamic_node_feats_dim}') # 5
```

latest ▾

```
print(f'TimeDelta                : {dg.time_delta}') # TimeDelta('s', value=1)
print(f'Device                   : {dg.device}') # torch.device(cpu)

# We can move the graph to GPU
dg = dg.to('cuda')
print(f'Device                   : {dg.device}') # torch.device(cuda:0)
```

> **Note**: The number of nodes is computed as `max(node_ids) + 1`.
>
> **Note**: If the `DGraph` is empty, `start_time` and `end_time` are `None`.
>
> **Note**: `len()` returns the number of timestamps (not the number of events) in the graph.

### Slicing: Creating new views

You can create a new `DGraph` view by slicing the underlying data. Currently, we support slicing by time, or by event index. Both operations are lightweight, as the storage is shared between `DGraph` instances. This makes it very fast to select subsets of your data.

You can slice temporal data using `slice_time()`. This returns a new `DGraph` containing only events within the specified time range (end time exclusive). Slicing is a lightweight operation since the underlying data storage is shared across `DGraph` instances.

> **Note**: These are both end-time *exclusive* operations.

Following from our previous code snippet:

```
sliced_dg = dg.slice_time(start_time=5, end_time=10)
print(sliced_dg.start_time) # 5
print(sliced_dg.end_time) # 9, end time exclusive
print(sliced_dg.num_edges) # 1
print(sliced_dg.device) # still on gpu
```

## 7. Materialization, Iteration and `DGBatch`

In practice, the typical workflow will require you to feed data into your model for training. For this purpose, we need to *materialize* the view.

The method on `DGraph` looks like:

```
def materialize(self, materialize_features: bool = True) -> DGB        ⑄ latest  ▼
    """Materialize the current DGraph slice into a dense `DGBat

    Args:
        materialize_features (bool, optional): If True, includes dynamic node
```

```
                features, node IDs/times, and edge features. Defaults to True.

        Returns:
            DGBatch: A batch containing src, dst, timestamps, and optionally
                features from the current slice.
        """
```

As described above, the output is a `DGBatch` object, which is nothing but a container of tensors corresponding to the materialized data of the `DGraph`, on device. By default, the `DGBatch` contains the following attributes:

```
@dataclass
class DGBatch:
    """Container for a batch of events/materialized data from a DGraph.

    Each `DGBatch` holds edge and node information for a slice of a dynamic graph,
    including optional dynamic node features and edge features. Hooks read and
write
    additional attributes to the container transparently during dataloading.

    Args:
        src (Tensor): Source node indices for edges in the batch. Shape `(E,)`.
        dst (Tensor): Destination node indices for edges in the batch. Shape
`(E,)`.
        time (Tensor): Timestamps of each edge event. Shape `(E,)`.
        dynamic_node_feats (Tensor | None, optional): Dynamic node features for
nodes
            in the batch. Typically sparse tensor of shape `(T x V x
d_node_dynamic)`.
        edge_feats (Tensor | None, optional): Edge features for the batch.
Typically
            sparse tensor of shape `(E x d_edge)` or `(T x V x V x d_edge)`
depending
            on storage.
        node_times (Tensor | None, optional): Timestamps corresponding to dynamic
node features.
        node_ids (Tensor | None, optional): Node IDs corresponding to dynamic node
features.
    """
```

For example:

```
# Our full graph view
dg_batch = dg.materialize(materialize_features=False) # Skip features
print(dg_batch.src) # torch.tensor([2, 2, 1], dtype=torch.long,
print(dg_batch.edge_feats) # None, because we skipped materiali

# Our sliced graph view (from start_time=5, end_time=10)
sliced_dg_batch = sliced_dg.materialize()
```

```
print(dg_batch.src) # torch.tensor([5], dtype=torch.long, device='cuda:0')
print(dg_batch.edge_feats is None) # False, we matrialized our slice of edge
features
```

> **Note**: Materializing a full graph view with features could be expensive, especially on large graphs. **Note**: The device of `DGraph` determines the device on which the `DGBatch` tensors are allocated.

## DGDataLoader

Internally, the `DGDataLoader` is responsible for materializing slices of graph data, using exactly the mechanics describe above. In particular, when you do something like:

```
from tgm import DGraph
from tgm.loader import DGDataLoader

dg = DGraph(...)
loader = DGDataLoader(dg, ...)

for batch in loader:
    ...
```

the data loader computes offsets into the storage, performs slicing operations, materializes the sliced views, and the applies hooks on the materialized data. See our hook management tutorial for more details.

---

## Summary

We learned about how `DGData` is used for loading data and preprocessing. We discussed how to created data splits and discretize your dataset to coarser time granularities. Once your data is loaded, you cast to `DGraph`, which is an immutable view of a slice of data.We showed how to query various attributes from a `DGraph`, and how to slice the `DGraph` in temporal snapshots. Finally, we showed how to *materialize* the data in `DGBatch` for training, and how the `DGDataLoader` does this internally during iteration.

With this foundation, you're ready to explore hook management and get started with our examples. Please feel free to reach out to us if anything is unclear or unintuitive. We are happy to discuss and improve your experience with TGM.

# Time Management in Temporal Graphs

This tutorial explains how **time deltas** work in TGM, how they influence graph construction, iteration, and discretization, and how to use them effectively.

---

## 1. TimeDeltaDG: High-Level Concept

A `TimeDeltaDG` defines the **temporal granularity** of a dynamic graph. It specifies the "unit of time" at which events (edges or nodes) are recorded. Think of it as the resolution of your graph's timeline.

See `tgm.timedelta.TimeDeltaDG` for full reference.

### Construction

You can create a `TimeDeltaDG` using a string alias or by explicitly providing a unit and a multiplier:

```
from tgm.timedelta import TimeDeltaDG

# Basic usage
td_seconds = TimeDeltaDG("s")        # 1-second granularity
td_days = TimeDeltaDG("D")           # 1-day granularity
td_biweekly = TimeDeltaDG("W", 2)    # 2-week (bi-weekly) granularity
```

**Event-Ordered vs. Time-Ordered**

There are 2 broad classes of `TimeDeltaDG` which determine how timestamps on a graph are interpreted:

- Event-Ordered ( `r` ): Events are only guaranteed to have a relative order. No real-world time unit is associated.
- Time-Ordered (e.g. second-wise ( `s` ), or daily ( `D` )): Standard time units like seconds, minutes, days, etc. Can perform coarsening or time conversion.

```
td_ordered = TimeDeltaDG("r") # Only relative order matters
```

The full list of time-ordered units is given below:

| Time Unit | Meaning |
| --- | --- |
| "Y" | Yearly |
| "M" | Monthly |
| "W" | Weekly |
| "D" | Daily |
| "h" | Hourly |
| "m" | Minute-wise |
| "s" | Second-wise |
| "ms" | Millisecond-wise |
| "us" | Microsecond-wise |
| "ns" | Nanosecond-wise |

**Coarser vs. Finer Granularities**

*Coarser* time granularities use lower resolution time units (e.g. week is coarser than day):

```
td_day = TimeDeltaDG("D")
td_week = TimeDeltaDG("W")
td_biweek = TimeDeltaDG("W", 2)
td_month = TimeDeltaDG("M")

print(td_week.is_coarser_than(td_day)) # True
print(td_biweek.is_coarser_than(td_week)) # True
print(td_month.is_coarser_than(td_biweek)) # True
```

> **Note**: Checking whether an event-ordered time delta is coarser or finer than an non-ordered is
> undefined and will raise a `EventOrderedConversionError`.

## 2. DGData Construction

Every `DGData` requires an associated `TimeDeltaDG`. Predefined datasets (e.g. `tgbl-wiki`) have native time deltas, usually in seconds.

If you are using a custom dataset, you must specify a time delta. If the exact temporal unit is unknown, you can resort to event-ordered granularity `TimeDelta('r')`, which is the default:

```python
from tgm.data import DGData

# Custom dataset with day granularity
dg_data = DGData(
    time_delta="D",
    timestamps=timestamps,
    ...
)

# Ordered dataset (relative order only)
dg_event_ordered = DGData(
    time_delta="r",
    timestamps=timestamps,
    ...
)
```

See `tgm.data.DGData` for full reference.

# 3. Temporal Data Iteration

The time delta also informs how you iterate over the graph. In this respect, the `DGDataLoader` uses two key parameters:

- `batch_unit`: Unit of time for batching (`r`, `D`, `h`)
- `batch_size`: Number of units or events per batch.

## Iteration Modes

There are two different modes of iteration in TGM, depending on whether the `batch_unit` parameter is event-ordered or time-ordered:

| Iteration Mode | Meaning | Example | Requires Time-Ordered Graph TimeDelta | Can produce empty batches |
|---|---|---|---|---|
| By Events (Event-Ordered) | Iterates over a fixed number of events at a time | Batch unit = `r` and batch size `N` yields N events per batch | No | No |
| By Time (Time-Ordered) | Iterates over a time window | Batch unit = `h` and batch size `3` yields 3 hours of data per batch | Yes | Yes |

**Note**: Time-based iteration can result in empty batches if no edge and no node events occur in the window. You can specify `on_empty='raise'` to error on empty batches, `on_empty='skip'` to ignore them, or `on_empty=None` to materialize the empty snapshots for your model. The default will materialize empty snapshots.

```
from tgm.loader import DGDataLoader

# Event-ordered iteration: yield 10 events per batch
loader = DGDataLoader(dg_data, batch_size=10)

# Time-ordered iteration: yield 3 days of data per batch, skip empty batches
loader_time = DGDataLoader(dg_data, batch_size=3, batch_unit='D', on_empty='skip')

# Time-ordered iteration: yield 3 days of data per batch, raise ValueError on
empty batches
loader_time = DGDataLoader(dg_data, batch_size=3, batch_unit='D',
on_empty='raise')
```

See `tgm.loader.DGDataLoader` for full reference. See `tgm.loader.DGDataLoader` for full reference.

## 4. Discretization: Coarsening Graphs

Discretization allows you to *coarsen* a time-ordered graph to a new time gr⸻ ⑂ latest ▼

- multiple edge and node events are partitioned into time buckets based on the requested granularity

- if multiple events map to the same edge in the same bucket, only the first occurence is kept (future versions will support other reduction ops)

This is useful for tuning dataset granularity (e.g. converting from continuous to discrete temporal graphs).

```
dg_data_second_wise = DGData.from_raw(
    time_delta="s",
    edge_timestamps=torch.tensor([15, 30, 45, 60]),
    edge_index=torch.tensor([[0, 1], [2, 3], [0, 1], [0, 1]),
    edge_feats=torch.tensor([[100, 200, 300, 400]]),
)

# Discretize from second-wise to minutely data
dg_data_minute_wise = dg_data_second_wise.discretize(time_delta="m",
reduce_op="first")

# After discretizing, note that the edge interaction between node 0 and 1 at time
15 and 45 are duplicates
# after grouping to minute-wise buckets (minute 0). In this case, we keep the
first event (edge_feats 100)
# and drop the second event (edge_feats 400).
print(dg_data.time_delta) # TimeDeltaDG("m")
print(dg_data.edge_timestamps) # torch.tensor([0, 0, 1])
print(dg_data.edge_index) # torch.tensor([[0, 1], [2, 3], [0, 1])
print(dg_data.edge_feats) # torch.tensor([[[100, 200, 400]])
```

> **Note**: Discretization is only defined for time-ordered graphs. Attempting to discretize an even-ordered `DGData` is undefined and will raise `InvalidDiscretizationError`.

## 5. Workflows

### TGB Datasets, Continuous-Time Temporal Graph Model

This is the simplest setup. Simply use `DGData.from_tgb()` to load the TGB dataset with its native time granularity. By default, `batch_unit='r'` in the data loader so we can iterate by batches of 200 events with:

```
from tgm import DGData, DGraph
from tgm.loader import DGDataLoader

data = DGData.from_tgb('tgbl-wiki')
dg = DGraph(data)
loader = DGDataLoader(dg, batch_size=200)
```

latest ▼

## TGB Datasets, Discrete-Time Temporal Graph Model

In this case, we can still load the native time granularity for the given TGB dataset. However, we need to specify a valid `batch_unit` in our dataloader. Recall, that internally, this applies a `TimeDeltaDG` conversion, and therefore, our iterating batch unit must be coarser (or the same granularity) as the underlying graph time unit.

Here, we use `tgbl-wiki` which has second-wise data, and we iterate over it in weekly snapshots:

```
from tgm import DGData, DGraph
from tgm.loader import DGDataLoader

data = DGData.from_tgb('tgbl-wiki')
dg = DGraph(data)
loader = DGDataLoader(dg, batch_unit='W')
```

We can just as easily iterate over biweekly graph snapshots:

```
from tgm import DGData, DGraph
from tgm.loader import DGDataLoader

data = DGData.from_tgb('tgbl-wiki')
dg = DGraph(data)
loader = DGDataLoader(dg, batch_unit='W', batch_size=2)
```

## Custom Datasets with Known TimeDelta

When working with custom datasets, it's likely that you have an underlying time granularity as determined by your data feed. For instance, you may be streaming log events with unix timestamps, or have pre-aggregated data arriving daily from a cron job.

In this case pretty much the same workflow as above can be used. Just make sure to pass the right unit when constructing your `DGData.from_raw()`. You may also be interested in discretizing your dataset into various granularities, and running some data analysis on the underlying graphs (e.g. figuring out number of nodes, edges, connected components etc).

## Custom Datasets with Unknown TimeDelta

It could occur that the underlying source time unit is not known a priori. In this situation, you can use the even-ordered time unit `TimeDeltaDG('r')` which preserves the rel[...]

without assuming a specific time unit.

latest

# Hook Management in TGM

Temporal graph learning pipelines often require dynamic transformations on graph batches—like sampling neighbors, generating negative edges, or moving data to GPU. TGM defines `DGHook` s to provide a flexible, composable way to perform these transformations automatically during batch iteration. Think of `DGHook` s as all the necessary data processing and operations before you feed the current batch into the TG ML model.

---

## 1. Hooks: The Basics

A `DGHook` is a callable object that takes a `DGBatch` (a batch of graph events) and a `DGraph` (a temporal view over the entire graph) as inputs and returns a transformed `DGBatch`, with additional properties.

See `tgm.graph.DGBatch` for a full reference of the base `DGBatch` yielded by our `DGDataLoader`.

Hooks declare the following information

- `requires: Set[str]` : Names of attributes that the hook needs to exist on the batch
- `produces: Set[str]` : Names of attributes from the batch that the hook requires
- `has_state: bool` : A flag to denote whether the hook stores state internally (i.e. some memory or attribute that may change upon subsequent invocations of the hook). An example of a stateful hook is a `RecencyNeighborSampler` which keeps track of node interactions over subsequent `__call__` s.

> Note: - `StatelessHook` : only transforms the batch, no internal state ( `has_state = False` ) -
> `StatefulHook` : maintains internal state, ( `has_state = True` )

### Built-in Hooks

TGM implements several commonly used hooks. The table below summarizes them:

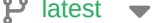 latest

| Hook Name | Type | requires | produces | Description |
|---|---|---|---|---|
| `NegativeEdgeSamplerHook` | Stateless | None | `neg`, `neg_time` | Generates random negatives for link prediction |
| `TGBNegativeEdgeSamplerHook` | Stateless | None | `neg`, `neg_time`, `neg_batch_list` | Loads pre-computed negative edges for TGB datasets |
| `NeighborSamplerHook` | Stateless | None | `nbr_nids`, `nbr_times`, `nbr_feats` | Uniform sampler neighbor for a given number of hops |
| `RecencyNeighborSamplerHook` | Stateful | None | `nbr_nids`, `nbr_times`, `nbr_feats` | Recency neighbor sampler for a given number of hops |
| `PinMemoryHook` | Stateless | None | None | Pins all `torch.Tensor` in `DGBatch` for fast CPU-GPU transfer |
| `DeduplicationHook` | Stateful | None | `unique_nids`, `global_to_local` | Computes unique node ids in `DGBatch` and a mapping from global (graph) to local (batch) coordinates |

## Custom Hooks

Along with the hooks provided by `TGM` team, users are welcome to write custom hooks to perform any operations on `DGBatch` as desired. For instance, if you are developing a new model or new sampling strategy, chances are, all you need to do is define a custom hook. The first step is to think about whether you need internal state. If not, you can subclass `tgm.hooks.StatelessHook`

For example, the following shows a simple implementation of a negative sampler random negative nodes in the range `[10, 100)`, and a corresponding *negative time* which matches the ground truth batch time:

```
from tgm.hooks import StatelessHook
from tgm import DGBatch, DGraph

class MyNegativeHook(StatelessHook):
    produces = {'my_neg', 'my_neg_time'}
    requires = set()

    def __call__(self, dg: DGraph, batch: DGBatch) -> DGBatch:
        batch.my_neg = torch.randint(10, 100, (len(batch.dst),))
        batch.my_neg_time = batch.time.clone()
        return batch
```

> **Important**: Each hooks adds attributes to the batch. Hooks that run after it may depend on these attributes (defined in `requires`). More on that later.

## 2. HookManager: Orchestrator of Hooks

Typically, a full training and evaluation pipeline will require multiple hooks, perhaps some of which execute conditionally on your workload (e.g. validation vs. test). The `HookManager` manages which hooks are applied to a batch, and in what order. You can think of it like a key-value store where:

- *Keys*: e.g. `'train'`, `'val'`, `'test'`
- *Values*: List of hooks associated with each key

Hooks are executed automatically during data loading, allowing different transformations to occur for different data splits. For instance:

```
from tgm.hooks import NegativeEdgeSamplerHook # A real negative edge sampler
from tgm.loader import DGDataLoader

# Create our graph
train_dg, test_dg = ...

# Initialize a hook manager with 'train' and 'test' keys
hm = HookManager(keys=['train', 'test'])

# Train: Random negatives
hm.register('train', NegativeEdgeSamplerHook(low=0, high=dg.num_nodes))

# Test: Use the dummy class we defined above
hm.register('test', MyNegativeHook())

train_loader = DGDataLoader(train_dg, hook_manager=hm)
test_loader = DGDataLoader(test_dg, hook_manager=hm)
```

latest ▼

**Important**: When creating custom hooks, you need to make sure you follow the correct hook API. See `tgm.hooks` for more information. A `BadHookProtocolError` will be thrown if you accidentlly tried registering a hook with the wrong API. We suggest you write some unit tests to accompany your custom protocols. You can see some of our hook tests as a starting point. If your hook has general utility to the TG community, we can add it to TGM and enable code re-use for other practitioners.

What now? Well, when we iterate our training graph, we have access to the attributes produced by `NegativeEdgeSamplerHook`, which are `neg` and `neg_time`. In order to see these transformations get applied, we need to *activate* the key we are interested in...

## 3. Context Management

In the previous section, we created a hook manager and added a hook to the 'train' key and another to the 'test' key. If we just try iterating the data, we won't see the attributes we want:

```
for batch in train_loader:
    assert batch.dst.shape() == batch.neg.shape() # AttributeError! No attribute
`neg` in batch

for batch in test_loader:
    assert batch.dst.shape() == batch.my_neg.shape() # AttributeError! No
attribute `my_neg` in batch
```

What we have to do is *activate* the keys we want. This allows us to selectively execute the right transformation, depending on which key is active. We can use the `with hm.activate()` context manager to do so:

```
with hm.activate('train'):
    for batch in train_loader:
        assert batch.dst.shape() == batch.neg.shape() # True

with hm.activate('test'):
    for batch in test_loader:
        assert batch.dst.shape() == batch.my_neg.shape() # True
        assert torch.all(batch.my_neg >= 10) # True
        assert torch.all(batch.my_neg < 100) # True
        assert torch.equal(batch.my_neg_time, batch.time) # True
```

> **Note**: The context manager is just syntactical sugar for the following:

```
with hm.activate(key):
    ...
```

```
#### Equivalent to
hm.set_active_hooks(key)
...
hm.set_active_hooks(None)
```

See `tgm.hooks.HookManager` for a full reference.

## State Reset

Often it will happen that hooks with internal memory (stateful hooks) require that some memory is reset, at an end of epoch, for instance. The `HookManager` will automatically walk through all the stateful hooks and call `reset_state()` internally when you issue:

```
hm.reset_state()
```

You can also selectively reset hooks for a particular key.

```
hm.reset_state('train')
```

# 4. Shared Hooks

In temporal graph learning, it is common that information you received in the past needed to be used for future prediction. For example, the stored neighbours in the `tgm.hooks.RecencyNeighborSampler` hook is state that must be carried to the validation phase to ensure that the models can access information from the training set. Therefore, this raises the need for sharing hook state of a hook across splits.

For this purpose, we have the notion of `shared hooks`, which are automatically attributed to **all** keys in the `HookManager`:

```
from tgm.loader import DGDataLoader

# Create our graph
train_dg, test_dg = ...

# Initialize a hook manager with 'train' and 'test' keys
hm = HookManager(keys=['train', 'test'])

# Register our dummy hook across both the train and test split
hm.register_shared(MyNegativeHook())
```

> *Note*: Using shared hooks is typically only useful if the hook has state, that needs to be shared across activation keys.

## 5. Hook Resolution

As you may have guessed, hooks add attributes that may depend on other hooks. Formally, the set of `requires` and `produces` attributes defined on `DGBatch` by the list of hooks defines a directed-acyclic-graph (*DAG*) for every key in the hook manager. When we activate a key, the hook manager performs a topological sort of the hook list and finds a topological ordering to execute during data loading. This is only done once and cached, until (if) you decide to add more hooks for that key.

The upside is that you shouldn't care what order you register your hooks in, the manager will figure it out. But, it's possible that no valid ordering exists.

For instance, suppose in our dummy hook, we added a requirement that our hook `requires` the batch attribute `foo`:

```python
from tgm.hooks import StatelessHook
from tgm import DGBatch, DGraph

class MyNegativeHookWithFoo(StatelessHook):
    produces = {'my_neg', 'my_neg_time'}
    requires = {'foo'} # This hook depends on batch.foo existing!

    def __call__(self, dg: DGraph, batch: DGBatch) -> DGBatch:
        batch.my_neg = torch.randint(10, 100, (len(batch.dst),))
        batch.my_neg_time = batch.time.clone()
        return batch
```

Now, if we register our hook and try to activate a key that uses it, we'll encounter the `tgm.UnresolvableHookDependenciesError`:

```python
# Register MyNegativeHook on 'train' then activate it and try iterating the data,
as before
hm.register('train', MyNegativeHookWithFoo()) # Ok, registered

with hm.activate('train'): # Raises tgm.UnresolvableHookDependenciesError
    ...
```

You will see the error message tell you that the manager could not find a valid ordering of hooks, and that's because no hook *produces* `'foo'`. If you encounter this, chances are you just misspelled either your `requires` or `produces` specification.

> *Note*: You can also manually try to resolve hooks for a specific key witho̶u̶t̶ ̶a̶c̶t̶i̶v̶a̶t̶i̶n̶g̶ ̶a̶n̶y̶t̶h̶i̶n̶g̶

```python
hm.resolve_hooks('train') # Raises tgm.UnresolvableHookDepenenciesError
```

You can inspect the resolved hooks according to the `__str__` method on the `HookManager`, to validate that everything is as expected as well:

```
print(hm)
```

It might give you something along the lines of:

```
HookManager:
  Shared hooks:
    - DeduplicationHook (requires=set(), produces={'unique_nids',
'global_to_local'})
    - MockHook (requires=set(), produces=set())
  Active key: None
  Keyed hooks:
    train:
      - DeduplicationHook (requires=set(), produces={'unique_nids',
'global_to_local'})
      - MockHook (requires=set(), produces=set())
      - MockHookRequires (requires={'foo'}, produces=set())
      - MockHookWithState (requires=set(), produces=set())
    val:
      - DeduplicationHook (requires=set(), produces={'unique_nids',
'global_to_local'})
      - MockHook (requires=set(), produces=set())
      - MockHookRequires (requires={'foo'}, produces=set())
```

# 6. Recipes

`TGM` offer a convenient way to setup common `HookManager` configuration by using `RecipeRegistry.build()` with a pre-defined recipe. For example, in the TGB `linkproppred` setting, the `HookManager` must register train, validation, and test hooks as follows:

```
dataset = PyGLinkPropPredDataset(
    name=dataset_name, root='datasets'
)

dataset.load_val_ns()
dataset.load_test_ns()
_, dst, _ = train_dg.edges
neg_sampler = dataset.negative_sampler

hm = HookManager(keys=['train', 'val', 'test'])
hm.register(
    'train', NegativeEdgeSamplerHook(low=int(dst.min()), high=int(dst.max()))
)
```

```
hm.register('val', TGBNegativeEdgeSamplerHook(neg_sampler, split_mode='val'))
hm.register('test', TGBNegativeEdgeSamplerHook(neg_sampler, split_mode='test'))
```

To minimize boilerplate and avoid accidental typos in this setup process, this procedure can be encapsulated in a function and registered through `RecipeRegistry` as follows:

```
@RecipeRegistry.register(RECIPE_TGB_LINK_PRED)
def build_tgb_link_pred(dataset_name: str, train_dg: DGraph) -> HookManager:
    try:
        from tgb.linkproppred.dataset_pyg import PyGLinkPropPredDataset
    except ImportError:
        raise ImportError('TGB required to load TGB data, try `pip install py-tgb`')

    dataset = PyGLinkPropPredDataset(
        name=dataset_name, root='datasets'
    )
    dataset.load_val_ns()
    dataset.load_test_ns()
    _, dst, _ = train_dg.edges
    neg_sampler = dataset.negative_sampler

    hm = HookManager(keys=['train', 'val', 'test'])
    hm.register(
        'train', NegativeEdgeSamplerHook(low=int(dst.min()), high=int(dst.max()))
    )
    hm.register('val', TGBNegativeEdgeSamplerHook(neg_sampler, split_mode='val'))
    hm.register('test', TGBNegativeEdgeSamplerHook(neg_sampler, split_mode='test'))

    return hm
```

`build_tgb_link_pred()` encapsulates procedure to set up `HookManager` for `TGB` linkpropred experiments and is registered to `RecipeRegistry` with the name defined by constant `RECIPE_TGB_LINK_PRED` as follows:

```
@RecipeRegistry.register(RECIPE_TGB_LINK_PRED)
```

Therefore, all we need to do to set up `HookManager` for `TGB` linkproppred is:

```
hm = RecipeRegistry.build(
    RECIPE_TGB_LINK_PRED, dataset_name=args.dataset, train_dg=train_dg
)
registered_keys = hm.keys
train_key, val_key, test_key = registered_keys
```

`TGM` team provided the implementation of recipe for `TGB` linkproppred, users are welcome to define their own `Recipe` , register it and build it with `RecipeRegistry.build()` .

## Summary

`DGHook` s are modular transformation applied to batches under the hood during data loading. The `HookManager` orchestrates hooks by key-value pair, and ensures correct execution order given the set of `requires` and `produces` attributes. After activating a given key, the yielded batch from the dataloader will have all the `produces` attributes computed for you.

By sub-classing either the `StatefulHook` or `StatelessHook` , you can define you own custom hooks in `TGM` .