# OpenReview forum: "TGM: A Modular and Efficient Library for Machine Learning on Temporal Graphs"
_ICLR.cc/2026/Conference — ICLR 2026 Poster_

### Official Review · Reviewer_pG5Q · 2025-10-25

**Soundness:** 3
**Presentation:** 2
**Contribution:** 4
**Rating:** 6
**Confidence:** 4

**Summary:**

This paper introduces a new library for temporal graph learning that unifies CTDG and DTDG, supporting a wide range of tasks and methods efficiently. The work includes several novel components, notably the unification of CTDG and DTDG and the hook management mechanism for obtaining training features. Its modular design is well-conceived and will allow researchers to easily develop, extend, and evaluate new methods.

**Strengths:**

The proposed library is timely and represents a valuable contribution to the field of temporal graph learning. It offers a convenient experimental infrastructure and a fair evaluation platform that can accelerate research progress. Compared with existing libraries, it appears more general and efficient, and it provides a diverse collection of datasets and methods to facilitate easy and consistent comparison across studies.

**Weaknesses:**

W1. My primary concern lies in the writing. First, while the authors emphasize the efficiency of TGM, the paper lacks a detailed explanation or analysis clarifying why TGM is more efficient than existing methods. Second, Section 3 spans nearly two pages but presents multiple items in a disconnected manner, making it difficult to follow the logical flow of ideas.

W2. I also noticed that a prior study, TGB-Seq, introduced several new datasets and integrated them into DyGLib. It would be better if the authors could incorporate the TGB-Seq datasets into TGM as well, since these datasets have been widely adopted in recent research.

**Questions:**

See weaknesses.

---

> ### Author Response · Authors · 2025-11-22
> **Author Response to Reviewer pG5Q**
>
> We thank the reviewer for taking the time to assess our paper and for the constructive suggestions. It's rewarding to see the recognition of key aspects of our work, e.g., that. offers a convenient and efficient experimental infrastructure with a diverse collection of datasets and methods, thereby contributing to the acceleration of research progress. In the following, we will reply to the reviewers' questions/comments and have revised our manuscript accordingly, with changes highlighted in blue.
>
> ---
>
> **W1.concern lies in the writing**
>
> > My primary concern lies in the writing. First, while the authors emphasize the efficiency of TGM, the paper lacks a detailed explanation or analysis clarifying why TGM is more efficient than existing methods. Second, Section 3 spans nearly two pages but presents multiple items in a disconnected manner, making it difficult to follow the logical flow of ideas.
>
>
> **Response:**
> Thank you for raising this point. We have refined and polished Section 3 accordingly, and added sub-headings to improve readability (all changes marked in blue). The first subsection now provides a structured walkthrough of our temporal graph formulation, beginning with the fundamental units of information (node and edge events) and culminating in our iteration and discretization operators. The second subsection then introduces our hook formulation, which directly informs the software design presented in Section 4.
>
> On the efficiency question, TGM achieves high efficiency through several key design choices:
>
> - **Circular buffers for neighbor sampling.** TGM uses circular buffers to store recent neighbors for each node directly on-device. This enables efficient updates as new edges arrive (without copying or reallocating memory) and cache-friendly reads of neighbor information (by the model), both of which are in the critical path of TGL training and inference workloads.
>
> - **Vectorized discretization.** Our graph discretization operator is fully vectorized in torch-native code, bypassing Python overhead and leveraging optimized kernels for operations edge reductions and feature transformations. As a result, discretization scales efficiently to large graphs and maintains compatibility with GPU acceleration.
>
> - **Immutable data assumptions.** TGM treats the underlying graph storage as immutable after ingestion. This guarantees that previously computed results (e.g., graph properties, neighborhood information, time queries) remain valid, allowing heavy use of caching and avoiding redundant recomputation across epochs.
>
> - **DGraph views over underlying storage.** Instead of copying or duplicating subgraphs, TGM provides lightweight temporal views that reference the original storage. This ensures that multiple temporal snapshots can be created with minimal memory overhead and tensor re-allocation.
>
> - **COO-based indexing.** Edges are indexed in time-sorted COO (coordinate) format, which enables binary search over timestamps and direct event-based filtering. This drives fast subgraph extraction during dataloading, leaving neighborhood extraction to online computation via our hook mechanism.
>
> **W2. TGB-Seq**
>
> > I also noticed that a prior study, TGB-Seq, introduced several new datasets and integrated them into DyGLib. It would be better if the authors could incorporate the TGB-Seq datasets into TGM as well, since these datasets have been widely adopted in recent research.
>
>
> **Response:**
> Thank you for this suggestion. Indeed, we agree that support for TGB-seq will be beneficial for the community and it has been widely-used in recent research. Therefore, we will incorporate native IO support for TGB-seq in the next release of TGM. In the meantime, we have implemented examples of running TGB-seq datasets in TGM and we report the TGM training latencies (per epoch) of TGN and TGAT in seconds.
>
> | Dataset      | TGN Latency | TGAT Latency |
> |--------------|-------------|--------------|
> | GoogleLocal  | 456.214     | 170.978      |
> | YouTube      | 868.003     | 291.157      |
> | Flickr       | 1792.323    | 1002.196     |

---

> > ### Comment · Reviewer_pG5Q · 2025-11-26
> >
> > Thank you for the detailed response — it addresses most of my concerns. I have one remaining question:
> >
> > Does TGM require more CPU memory than other existing libraries, given that it makes heavy use of caching?

---

> > > ### Author Response · Authors · 2025-11-27
> > > **Author Response to Reviewer pG5Q**
> > >
> > > Thank you for your follow-up question and we are glad to have addressed most of your concerns. We would like to answer your question below.
> > >
> > > ---
> > >
> > > **CPU memory of TGM**
> > >
> > > > Does TGM require more CPU memory than other existing libraries, given that it makes heavy use of caching?
> > >
> > > **Response**: Thank you for the question. In the following table, we compare with DyGLib on RAM and GPU usage (the lowest usage is highlighted in bold).
> > >
> > > **Memory (in MB) Comparison: TGM (ours) vs. DyGLib on tgbl-wiki**
> > > | Metric     | DyGLib  | TGM  |
> > > |------------|-------------|----------|
> > > | Peak RAM   | 1028.38     | **421.89**   |
> > > | Peak GPU   | **1144.54**     | 1272.30  |
> > >
> > > **Memory (in MB) Comparison: TGM (ours) vs. DyGLib on tgbl-subreddit**
> > > | Metric     | DyGLib | TGM |
> > > |------------|-------------|----------|
> > > | Peak RAM   | 4360.27     | **1729.95**  |
> > > | Peak GPU   | 1482.54     | **1294.15**  |
> > >
> > > **Memory (in MB) Comparison: TGM (ours) vs. DyGLib on tgbl-lastfm**
> > >
> > > | Metric     | DyGLib | TGM |
> > > |------------|-------------|----------|
> > > | Peak RAM   | 3764.99     | **3371.52**  |
> > > | Peak GPU   | **772.16**      | 819.59   |
> > >
> > > TGM consistently uses less peak RAM than DyGLib while maintaining comparable GPU memory usage. This efficiency comes from a few deliberate design choices: we mainly cache binary-search for temporal graph queries which consumes minimal memory, our graph views avoid redundant data copies during data loading; and we use only the necessary data types needed for timestamps and node IDs. Together, these choices keep memory overhead minimal while preserving speed.

---

### Official Review · Reviewer_KuR4 · 2025-10-30

**Soundness:** 3
**Presentation:** 3
**Contribution:** 2
**Rating:** 4
**Confidence:** 4

**Summary:**

This paper introduces TGM, a research-oriented library for Temporal Graph Learning (TGL)
that supports both continuous-time (CTDG) and discrete-time (DTDG) dynamic graphs within a single modular system.
The library proposes a formal unification of both paradigms via the notion of graph discretization and time granularity.
TGM supports link-, node-, and graph-level tasks and demostrantes significant efficiency gains over prior libraries such as DyGLib and UTG.

**Strengths:**

- The paper addresses a timely and relevant problem in dynamic graph learning, where current frameworks lag behind static counterparts (e.g., PyG, DGL) in terms of flexibility, modularity, and computational efficiency.
- The manuscript is well written and clearly structured, providing sufficient context and motivation for the proposed library.
- If properly released and maintained as open-source, TGM could serve as a standard reference library for temporal graph learning research.
- The framework demonstrates substantial efficiency gains across multiple models and tasks, indicating careful engineering and system design.

**Weaknesses:**

- The empirical section primarily focuses on efficiency metrics and new results on CTDG tasks solved using DTDG-based models. However, these results offer limited analysis of performance in comparison with prior libraries and existing literature.
- Some aspects are insufficiently detailed:
    1) It is not clear whether experiments across libraries use identical data splits, hyperparameters, preprocessing, etc.
    2) The paper does not clearly explain how batches are processed, specifically, whether events or snapshots within a batch are handled in parallel or sequentially, and if parallel processing is used, if temporal causality is preserved.
    3) The treatment of deletion events (e.g., node removal) is not detailed in the paper.
    4) It is unclear if irregularly sampled snapshot-based tasks (e.g., as in [4,5]) can be processed within TGM.
- TGM currently implements tasks for CTDGs and enables DTDG models to operate on these tasks. However, the TGL community would greatly benefit if TGM also supported tasks specifically designed for DTDGs, such as Metr-LA and Pems-Bay [6], making it a truly unified library that facilitates research across both CTDG and DTDG paradigms.
- When evaluating DTDG-based models, the authors should include comparisons against dedicated DTDG frameworks, such as Torch Spatiotemporal or PyTorch Geometric Temporal, to reduce the large number of unsupported baselines reported in Tables 3, 4, and 9.
- The unified conceptual view connecting CTDGs and DTDGs is interesting; however, the theoretical relationship between these two temporal graph paradigms has already been discussed in prior works, including [1, 2, 3].
  Although not implemented in a general-purpose library, these contributions should be properly acknowledged in the manuscript.
- Given the growing interest on long-range information propagation and oversquashing in temporal GNNs [7], the authors should consider including CTAN [8], a model specifically designed to address this problem, within the TGM. This addition would further broaden the TGM's coverage and enhance its utility for research on novel temporal GNN architectures.

-----

[1] [Representation Learning for Dynamic Graphs: A Survey. JMLR 2020](https://jmlr.csail.mit.edu/papers/volume21/19-447/19-447.pdf)

[2] [Deep learning for dynamic graphs: models and benchmarks. IEEE TNNLS 2024](https://ieeexplore.ieee.org/document/10490120)

[3] [Graph neural networks for temporal graphs: State of the art, open challenges, and opportunities. TMLR 2023](https://openreview.net/pdf?id=pHCdMat0gI)

[4] [Graph Neural Controlled Differential Equations for Traffic Forecasting. AAAI 2022](https://cdn.aaai.org/ojs/20587/20587-13-24600-1-2-20220628.pdf)

[5] [Temporal Graph ODEs for Irregularly-Sampled Time Series. IJCAI 2024](https://www.ijcai.org/proceedings/2024/0445.pdf)

[6] [Diffusion Convolutional Recurrent Neural Network: Data-Driven Traffic Forecasting. ICLR 2018](https://arxiv.org/abs/1707.01926)

[7] [Over-squashing in Spatiotemporal Graph Neural Networks. 2025](https://arxiv.org/pdf/2506.15507)

[8] [Long Range Propagation on Continuous-Time Dynamic Graphs. ICML 2024](https://openreview.net/pdf?id=gVg8V9isul)

**Questions:**

Refer to the paper’s weaknesses.

---

> ### Author Response · Authors · 2025-11-22
> **Author Response to Reviewer KuR4 (Part 1)**
>
> We thank the reviewer for taking the time to review our paper and for offering valuable suggestions. We are pleased that the reviewer recognized our TGM as a standard reference library for temporal graph learning research with careful engineering and system design and highlighted the contribution of our work in addressing a timely and relevant problem in dynamic graph learning: flexibility, modularity, and computational efficiency. In the following, we will reply to the reviewer's questions and comments and revise our manual scripts to incorporate the reviewer’s feedback in blue.
>
> ---
>
> **W1. Limited analysis of performance in comparison with prior libraries**
>
> > The empirical section primarily focuses on efficiency metrics and new results on CTDG tasks solved using DTDG-based models. However, these results offer limited analysis of performance in comparison with prior libraries and existing literature.
>
> **Response:**
> Thank you for this point. We would like to clarify that in the paper, we compare with efficient and widely-used TGL libraries such as DyGLib, TGLite and TGL often being competitive or better in efficiency (as seen in Table 3).
>
>  In terms of model performance, we report correctness results originally in Table 12 in Appendix B.4 (now we have moved to the Section 5.2 in the main paper and also attached below). Here, we report the model performance on MRR for the tgbl-wiki dataset from the widely-used TGB benchmark. TGM results largely align with the TGB leaderboard in terms of model ranking and performance without extensive hyperparameter tuning thus validating the correctness of TGM implementations.
>
> | **Category** | **Model**  | **tgbl-wiki Test MRR (↑)**  |
> |-------|--------|--------|
> | Baselines  | Edgebank    | 0.527   |
> | DTDG | GCN | 0.410 ± 0.019   |
> |   | GCLSTM | 0.364 ± 0.015 |
> | | TGCN| 0.332 ± 0.004 |
> | CTDG | TGAT  | 0.322 ± 0.013  |
> | | TGN   | 0.527 ± 0.008  |
> |   | GraphMixer  | 0.567 ± 0.018  |
> |  | DyGFormer   | 0.712 ± 0.009 |
> | | TPNet | 0.747 ± 0.037  |
>
> **W2.1. Experiments across libraries**
>
> > It is not clear whether experiments across libraries use identical data splits, hyperparameters, preprocessing, etc.
>
> **Response:**
> For experiments, we use the standard TGB data splits which uses 70/15/15 chronological splits for train, validation and test. For hyperparameters, we report all hyperparameters for all models in Table 14. To ensure fair efficiency comparison with prior libraries, experiments were run on Ubuntu 20.04 with 64 GB RAM, 4 isolated AMD EPYC 7502 CPU cores, and a single 80 GB A100 GPU. Jobs were managed with SLURM to ensure isolated
> environments and no concurrent interference. More experimental details are reported in Appendix F.  We have revised our manuscript to highlight the consistency in data splits, hyperparameters in all experiment at the beginning of Section 5.
>
> **W2.2. how batches are processed**
>
> > The paper does not clearly explain how batches are processed, specifically, whether events or snapshots within a batch are handled in parallel or sequentially, and if parallel processing is used, if temporal causality is preserved.
>
> **Response:** Thank you for the question. TGM preserves temporal causality by processing batches in chronological order. Within each batch, events are processed in parallel: TGM hooks such as neighbor sampling, device transfers, and negative edge generation operate concurrently on all events, contributing to overall efficiency.
>
> For TGNN models, embeddings and model state are updated only after processing each batch, ensuring that predictions use only information from prior events. Temporal leakage is further prevented by the recency neighbor sampling hook, which updates its internal state only after the batch is fully processed, maintaining correct temporal dependencies.
>
> **W2.3. Deletion events**
>
> > The treatment of deletion events (e.g., node removal) is not detailed in the paper.
>
> **Response:** Thank you for the question. Node deletion events are handled implicitly in the current TGM formulation. Most public datasets only record edge insertions, rarely deletions, and typically assume that edges disappear if they are not repeated in the next timestamp. TGM follows this convention and does not explicitly support node removal. In practice, a node is considered deleted if it has no node or edge events within the current batch.
>
> **W2.4. Irregularly sampled snapshot**
>
> > unclear if irregularly sampled snapshot(e.g., as in [4,5]) can be processed within TGM.
>
> **Response:**
> Thank you for the question. Irregularly sampled events are fully supported in TGM. We provide efficient time-slicing operations with arbitrary time windows via binary search and direct event indexing on the underlying graph storage. For example, you can query all events within a time interval with custom start and end time. Combining these query operations makes it straightforward to work with irregularly sampled snapshots, as in [4,5].

---

> > ### Author Response · Authors · 2025-11-22
> > **Author Response to Reviewer KuR4 (Part 2)**
> >
> > **W3.Metr-LA and Pems-Bay**
> >
> > > TGM currently implements tasks for CTDGs and enables DTDG models to operate on these tasks. However, the TGL community would greatly benefit if TGM also supported tasks specifically designed for DTDGs, such as Metr-LA and Pems-Bay [6]
> >
> > **Response:**
> > Thank you for your suggestion. Spatio-temporal signals can be modeled as dynamic node events in TGM, allowing the research flexibility to investigate more tasks on DTDGs. We have added the Metr-LA and Pems-Bay tasks as you suggested to TGM. The table below compares the efficiency of TGM with that of PyG Temporal in training latency for these tasks. The following table shows the training time per epoch in seconds with comparison between TGM and PyG Temporal.
> >
> > | Library | TGCN (METR_LA) | GCLSTM (METR_LA) | TGCN (PEMS_BAY) | GCLSTM (PEMS_BAY)|
> > |---------------|-----------|-----------|------------|------------|
> > | TGM (ours)    | 13.5070   | 17.1968   | 21.7525    | 26.6992    |
> > | PyG Temporal  | 14.8245   | 16.6192   | 23.2518    | 25.7316    |
> >
> > **W4.Comparisons against Torch Spatiotemporal or PyTorch Geometric Temporal**
> >
> > > When evaluating DTDG-based models, the authors should include comparisons against dedicated DTDG frameworks, such as Torch Spatiotemporal or PyTorch Geometric Temporal, to reduce the large number of unsupported baselines reported in Tables 3, 4, and 9.
> >
> > **Response:**
> > Thank you for the comment. In Tables 3, 4, and 9, a cross (×) indicates that a model is not supported in the corresponding library. In contrast, TGM supports all listed models, whereas DyGLib, TGLite, and TGL often lack support for many CTDG and DTDG models. This highlights TGM’s broad coverage across state-of-the-art temporal graph methods.
> >
> > As suggested, we have added efficiency comparisons with PyG Temporal on the METR_LA and PEMS_BAY tasks (see W3). TGM matches PyG Temporal in runtime while supporting more models (including SOTA CTDG methods) and additional tasks (link-, node-, and graph-level), demonstrating both comprehensiveness and efficiency.
> >
> > **W5.theoretical relationship between these two temporal graph paradigms**
> >
> > > The unified conceptual view connecting CTDGs and DTDGs is interesting; however, the theoretical relationship between these two temporal graph paradigms has already been discussed in prior works, including [1, 2, 3]. Contributions should be properly acknowledged in the manuscript.
> >
> > **Response:**
> > Thank you for the comment. We acknowledge prior works [1,2,3] that discuss the theoretical differences between CTDGs and DTDGs, namely that CTDGs operate on graph streams while DTDGs operate on graph snapshots. We have revised our our manuscript Section 2 (highlighted in blue) to cite and discuss these contributions.
> >
> > However, previous discussions mostly stop at conceptual differences and do not provide a practical framework for comparing or unifying the two paradigms. For instance, applying DTDG models to CTDG tasks requires forming discrete snapshots over continuous time, which is nontrivial. TGM addresses this by formalizing time granularity and enabling snapshot formation by grouping edges over fixed intervals. In essence, we view CTDGs and DTDGs as two iteration modes over the same underlying temporal graph, providing a unified conceptual and practical framework for analysis, benchmarking, and hybrid modeling.
> >
> > **W6.CTAN**
> >
> > > Given the growing interest on long-range information propagation and oversquashing in temporal GNNs [7], the authors should consider including CTAN [8], a model specifically designed to address this problem, within the TGM
> >
> > **Response:** Thank you for suggesting this model. We are currently implementing a working example of CTAN in TGM and will support it in TGM in the next release.

---

> ### Comment · Reviewer_KuR4 · 2025-11-25
>
> I thank the Authors for their response, which has clarified most of my concerns. Following this, and given the Authors' willingness to improve the coverage of their library in terms of models and datasets (as discussed also in other Reviewers' rebuttals), I am happy to increase my score. I would be willing to further increase it if authors can strengthen the coverage of TGM. Below, I provide follow-up comments for the rebuttal.
>
> **W2.2. how batches are processed**
>
> I thank the Authors for their clarification. However, I remain concerned about the implications of processing events within a batch in parallel.
> This design choice implicitly assumes independence among events in the same batch. While this improves computational efficiency, it also raises concerns regarding the preservation of more "fine-grained" temporal dependencies. In many temporal graph scenarios, events occurring close in time (and even within the same batching window) can have causal relationships or influence each other through rapid-evolving dynamics. By processing these events in parallel, the model may be unable to incorporate such immediate interactions, leading to potential information loss during embedding updates.
> I encourage the Authors to discuss the implications of this assumption more explicitly and to consider whether alternative batching strategies could mitigate this limitation.
>
> **W3.Metr-LA and Pems-Bay**
>
> I thank the Authors for including this new experiment.
> While the reported results provide an interesting perspective on runtime, the analysis again offers limited insight into relative performance compared with prior libraries. The current discussion emphasizes efficiency but does not clearly position TGM with respect to existing baselines in terms of predictive accuracy. A more comprehensive comparison would significantly help readers evaluate the benefits and limitations of the proposed library.

---

> > ### Author Response · Authors · 2025-11-27
> > **Author Response to Reviewer KuR4**
> >
> > Thank you for raising your score and we are glad to have addressed most of your concerns. We appreciate your constructive feedback and would like to address your comments below.
> >
> > ---
> >
> > **strengthen the coverage of TGM**
> >
> > > improve the coverage of their library in terms of models and datasets
> >
> > **Response** Based on your feedback and other reviewers’ suggestions in the rebuttal, we plan to support new models in TGM including EvolveGCNO and CTAN (increasing the model coverage to 11 models) as well as providing native IO support for TGB-seq and new graph property prediction datasets (increasing the out-of-the-box dataset coverage to 24 diverse temporal networks) in the next TGM release. To demonstrate even broader model coverage, we have also added an implementation of the widely-used ROLAND DTDG model [1] to TGM as well. We report its performance on the tgbl-wiki dataset below. We see that ROLAND performs the best in the DTDG model category thus making it a valuable addition to TGM.
> >
> > | **Category**  | **Model**  | **tgbl-wiki Test MRR (↑)** |
> > |------------|-------------|---------|
> > | Baselines  | Edgebank  | 0.527  |
> > | DTDG  | **ROLAND (new)**  | 0.464 ± 0.024     |
> > |            | GCN  | 0.410 ± 0.019     |
> > |            | GCLSTM   | 0.364 ± 0.015   |
> > |            | TGCN  | 0.332 ± 0.004     |
> > | CTDG | TGAT| 0.322 ± 0.013      |
> > |            | TGN | 0.527 ± 0.008         |
> > |            | GraphMixer | 0.567 ± 0.018   |
> > |            | DyGFormer   | 0.712 ± 0.009  |
> > |            | TPNet       | 0.747 ± 0.037   |
> >
> > All of the additions above have been implemented in pull requests, and we will add full test coverage of these new models and datasets prior to the next TGM release. TGM will continue to evolve with additional support and remain a reliable, open-source reference library for the community.
> >
> > [1] You, Jiaxuan, Tianyu Du, and Jure Leskovec. "ROLAND: graph learning framework for dynamic graphs." Proceedings of the 28th ACM SIGKDD conference on knowledge discovery and data mining. 2022.
> >
> > **W2.2. how batches are processed**
> >
> > >  I encourage the Authors to discuss the implications of this assumption more explicitly and to consider whether alternative batching strategies could mitigate this limitation.
> >
> > **Response.** Thank you for this question. Indeed as you mentioned, this is a known trade-off between efficiency and model update frequency in temporal graph learning. The larger the batch size, the more events are processed in parallel thus the more efficiently data is processed. On the contrary. smaller batches provide more fine-grained updates to the model thus allows the model to see more recent information and avoids the staleness problem. In our TGM experiments, we follow the  batch size of 200 edges as used in TGB evaluation for comparisons to prior work.
> >
> > We also agree that this implication is important and warrants deeper investigation as a future direction for TGM. Because TGM supports time operations and allows users to slice graphs by arbitrary start and end timestamps, exploring dynamic batch sizes is straightforward. We are open to further studying this idea and potentially implementing an optimal batching strategy that balances efficiency with update frequency. To clarify the implications of batching as you suggested, we have added a section discussing this topic in the Appendix of the revised paper.
> >
> > **W3.Metr-LA and Pems-Bay**
> >
> > >  The current discussion emphasizes efficiency but does not clearly position TGM with respect to existing baselines in terms of predictive accuracy. A more comprehensive comparison would significantly help readers evaluate the benefits and limitations of the proposed library.
> >
> > **Response**
> > Thank you for the question. We have added the performance results for T-GCN on both datasets, reproduced based on the official PyTorch Geometric Temporal [examples](https://github.com/benedekrozemberczki/pytorch_geometric_temporal/tree/master/examples/indexBatching/tgcn). The numbers correspond to mean absolute test-time error for 1D signal prediction after 30 training epochs with default hyperparameters. We report mean and standard deviations across 3 seeds. The implementation in TGM matches the performance of PyG Temporal as expected thus ensuring the correctness of our implementation.
> >
> >
> > | Method       | METR-LA Test MAE (mean ± std)   |
> > |--------------|-----------------------|
> > | pyg-temporal | 6.072 ± 0.013       |
> > | TGM          | 6.079 ± 0.024       |
> >
> > | Method       | PEMS-BAY Test MAE (mean ± std)   |
> > |--------------|-----------------------|
> > | pyg-temporal | 3.558 ± 0.003       |
> > | TGM          | 3.539 ± 0.019       |
> >
> > ---
> >
> > Thank you again for your engaged discussion. We hope our comments here have addressed your remaining concerns and we kindly hope you may consider reflecting this in your final assessment.

---

> > > ### Author Response · Authors · 2025-11-28
> > > **Official Comment by Authors**
> > >
> > > We thank the reviewer for raising their score from 4 to 6 following the rebuttal discussions.

---

### Official Review · Reviewer_SxD6 · 2025-10-31

**Soundness:** 2
**Presentation:** 3
**Contribution:** 3
**Rating:** 6
**Confidence:** 4

**Summary:**

This paper presents a research‑oriented software library that unifies continuous‑time and discrete‑time learning on temporal graphs under one formal and practical framework. The authors introduce a typed hook abstraction for composing common temporal operations, and they formalize time‑granularity conversion via a discretization operator, enabling iteration either by events or by time. The system follows a three‑layer architecture, supports node and edge events, and implements representative models spanning message‑passing, transformer‑based, and snapshot methods. Empirically, TGM reports stronger efficiency  than DyGLib and UTG. It also enables research case studies on dynamic graph properties, snapshot granularity, and batching effects.

**Strengths:**

S1. This paper is generally well-writen and easy to follow.

S2. This paper proposes a clean theoretical unification of CTDG and DTDG via the notion of a native time granularity and a principled discretization operator.

S3. This paper introduces a typed hook formalism with explicit dependency contracts, which makes complex temporal pipelines composable and easier to reason about and test.

S4. This paper provides a well‑architected system that separates data storage, execution, and model layers. the proposed framework demonstrates broad coverage of tasks and models.

**Weaknesses:**

W1. This paper’s primary contribution is a software framework. While the system is valuable, the methodological novelty is limited relative to prior unification attempts (e.g., UTG) and existing libraries (e.g., DyGLib), raising questions about the conceptual contributions beyond engineering improvements.

W2. This paper claims to unify CTDG and DTDG approaches, and it indeed provides a conceptual bridge through an event-sequence formulation and discretization operator. However, the empirical section does not yet demonstrate unification beyond efficiency. For example, there is no study comparing a CTDG model and a DTDG model on exactly the same data, tasks, and evaluation protocol.

W3. The coverage of DTDG methods in the library and experiments is limited. The implemented and evaluated snapshot-based models are mainly GCN, GCLSTM, and T-GCN, while stronger or more representative DTDG baselines such as EvolveGCN[1] and DySAT[2] are neither implemented nor evaluated.

W4. The experimental section focuses heavily on efficiency, but it does not verify correctness against the original authors’ official implementations or reported numbers across multiple metrics. As a result, the validity of the re-implemented models cannot be fully confirmed.

W5. This paper lacks empirical comparisons with PyG Temporal[3] and Torch Spatiotemporal[4], two DTDG-oriented libraries that are discussed in the related-work section but not benchmarked in Section 5.

Minor Typos.

+ "up to 246× than DyGLib" -> "up to 246× faster than DyGLib".

+ The spelling of behavior and behaviour should be made consistent.

+ Table 8 "first and second are highlighted" -> "First and second best are highlighted"



Reference

[1] EvolveGCN: Evolving Graph Convolutional Networks for Dynamic Graphs

[2] Dynamic Graph Representation Learning via Self-Attention Networks

[3] PyTorch Geometric Temporal: Spatiotemporal Signal Processing with Neural Machine Learning Models

[4] https://github.com/TorchSpatiotemporal/tsl

**Questions:**

1. Will the framework be adapted to text-attributed dynamic graphs[1]?

[1] DTGB: A Comprehensive Benchmark for Dynamic Text-Attributed Graphs

---

> ### Author Response · Authors · 2025-11-22
> **Author Response to Reviewer SxD6 (Part 1)**
>
> We thank the reviewer for taking the time to review our paper and for offering valuable suggestions. We are glad that the reviewer finds our manuscript well-written and easy to follow, and appreciates TGM as a robust system that supports a variety of research case studies on dynamic graphs. In the following, we will reply to the reviewer's questions and comments and revise our manual script to incorporate the reviewer’s feedback in blue.
>
> ---
>
> **W1.questions about the conceptual contributions beyond engineering improvements.**
>
> > This paper’s primary contribution is a software framework. While the system is valuable, the methodological novelty is limited relative to prior unification attempts (e.g., UTG) and existing libraries (e.g., DyGLib), raising questions about the conceptual contributions beyond engineering improvements.
>
> **Response:** Thank you for the thoughtful comment. While TGM has a valuable system contribution, we would like to emphasize that it also introduces new methodological abstractions that go beyond engineering improvements and are not present in prior unification attempts such as UTG or existing libraries like DyGLib.
>
> - **A true unification of CTDG and DTDG under one abstraction.** Prior works discuss CTDG and DTDG as categories, but existing libraries still treat them as separate modalities with incompatible data formats and tooling. TGM is the first to operationalize a single underlying temporal graph that can be iterated either by events (CTDG view) or by time intervals (DTDG view), controlled entirely by a user-specified time granularity. This makes it possible to: 1). switch between CTDG and DTDG representations without changing model code,
> 2). benchmark both families uniformly and 3). explore hybrid or adaptive time representations previously impractical due to fragmented tooling.
>
> - **A novel hook formalism tailored to temporal graphs.** TGM introduces a new abstraction (hooks) that standardize temporal graph operations while preventing temporal leakage(e.g., neighbor sampling, negative sampling). Unlike generic data transforms or message-passing primitives, hooks: 1). maintain state across time (critical for temporal dependencies), 2). enforce typed tensor I/O for safe composition of multi-step pipelines, and
> 3). support not only model training but also analytical workflows (e.g., density-of-states or spectral statistics).  This is a conceptual contribution aimed at structuring temporal graph workflows in a way that existing libraries do not.
>
> - **New capabilities not available in prior libraries.** TGM is the first temporal graph library to: 1). natively support dynamic node events (not just edge events), 2). make time granularity and time conversions first-class operators and 3). support link, node, and graph tasks within one system. Overall, our design introduces new abstractions and a genuine conceptual unification that we believe meaningfully advances the methodological foundations of temporal graph learning.
>
> **W2. demonstrate unification beyond efficiency**
>
> > However, the empirical section does not yet demonstrate unification beyond efficiency. For example, there is no study comparing a CTDG model and a DTDG model on exactly the same data, tasks, and evaluation protocol..
>
> **Response:** Thank you for the comment. We agree that demonstrating a unified empirical comparison is important. Due to space constraints, these results were originally placed in Appendix B.4 (Table 12), but we have now moved them into the main paper for clarity. In these experiments, we evaluate CTDG and DTDG models side-by-side on exactly the same data, task, and evaluation protocol, using two standard TGB benchmarks: link prediction on tgbl-wiki and node property prediction on tgbn-trade.
>
> To ensure fairness, all models use the same batch size (200 events) and follow TGB’s official evaluation procedure. TGM’s results closely follow the TGB leaderboard ordering without extensive hyperparameter tuning, demonstrating correctness.
>
> The unified comparison yields a clear pattern: CTDG models (e.g., TPNet, DyGFormer) perform best on event-intensive link prediction tasks such as Wikipedia, while DTDG models (e.g., GCN, GCLSTM) perform best on snapshot-based node property prediction tasks such as tgbn-trade.
>
> | **Category**  | **Model**  | **tgbl-wiki Test MRR (↑)** | **tgbn-trade Test NDCG (↑)**|
> |------------|-------------|---------|-----------|
> | Baselines  | Edgebank  | 0.527  | ---  |
> |        | P.F. | ---  | 0.855 |
> | DTDG  | GCN  | 0.410 ± 0.019     | 0.629 ± 0.009   |
> |            | GCLSTM   | 0.364 ± 0.015   | 0.692 ± 0.002     |
> |            | TGCN  | 0.332 ± 0.004     | 0.458 ± 0.007       |
> | CTDG | TGAT| 0.322 ± 0.013      | 0.309 ± 0.002   |
> |            | TGN | 0.527 ± 0.008         | 0.329 ± 0.002     |
> |            | GraphMixer | 0.567 ± 0.018       | ---        |
> |            | DyGFormer   | 0.712 ± 0.009  | 0.312 ± 0.0003   |
> |            | TPNet       | 0.747 ± 0.037   | ---       |

---

> > ### Author Response · Authors · 2025-11-22
> > **Author Response to Reviewer SxD6 (Part 2)**
> >
> > **W3.DTDG methods in the library is limited**
> >
> > > The coverage of DTDG methods in the library and experiments is limited. The implemented and evaluated snapshot-based models are mainly GCN, GCLSTM, and T-GCN, while stronger or more representative DTDG baselines such as EvolveGCN[1] and DySAT[2]
> >
> > **Response:** Thank you for the suggestions. We are actively implementing EvolveGCN in TGM and plan to include it in the next release. We plan to first support models with public results on TGB, which enables correctness verification. TGM is under continuous development, and we will expand coverage to include additional representative DTDG models such as DySAT in future updates.
> >
> > **W4.Correctness against official implementations**
> >
> > > The experimental section focuses heavily on efficiency, but it does not verify correctness against the original authors’ official implementations or reported numbers across multiple metrics. As a result, the validity of the re-implemented models cannot be fully confirmed.
> >
> > **Response:**
> > Thank you for the comment. In TGM, we verify correctness using the standard TGB evaluation procedure. These results, originally in Appendix B.4 (Table 12), are now moved to Section 5.2 of the main paper for visibility. TGM implementation largely matches the reported result on the TGB leaderboard thus confirming the validity of the implementation. Note that the TGM results are without extensive hyperparameter tuning when compared to the leaderboard ones. The correctness results are also shown in W2 earlier in the thread.
> >
> > **W5.Empirical comparisons with PyG Temporal and Torch Spatiotemporal**
> >
> > > This paper lacks empirical comparisons with PyG Temporal[3] and Torch Spatiotemporal[4], two DTDG-oriented libraries that are discussed in the related-work but not benchmarked in Section 5.
> >
> > **Response:** Thank you for the comment. TGM is highly flexible and readily supports spatio-temporal tasks. We have added examples of running PyG Temporal tasks within TGM, along with an efficiency benchmark. TGM matches PyG Temporal in runtime while additionally supporting more models (including state-of-the-art CTDG methods) and more tasks (link-, node-, and graph-level). This demonstrates that TGM provides a more comprehensive and unified framework for temporal graph learning. The following table shows the training time per epoch in seconds comparison between TGM and PyG Temporal.
> >
> > | Library | TGCN (METR_LA) | GCLSTM (METR_LA) | TGCN (PEMS_BAY) | GCLSTM (PEMS_BAY)|
> > |---------------|-----------|-----------|------------|------------|
> > | TGM (ours)    | 13.5070   | 17.1968   | 21.7525    | 26.6992    |
> > | PyG Temporal  | 14.8245   | 16.6192   | 23.2518    | 25.7316    |
> >
> > **Q1.Text-attributed dynamic graphs**
> >
> > > Will the framework be adapted to text-attributed dynamic graphs
> >
> > **Response:** Thank you for the question. TGM supports text-attributed dynamic graphs by treating text features as either dynamic node events (if changing) or static node features, and by incorporating edge text features as edge attributes. TGM provides a flexible custom data loader that allows users to add arbitrary node and edge features. We plan to expand support for text-attributed dynamic graph benchmarks, such as DTGB, in future releases.

---

> > > ### Comment · Reviewer_SxD6 · 2025-11-26
> > > **Response to authors**
> > >
> > > Thank you for your response. Your reply has solved the majority of my problems, so I will keep my score.

---

> > > > ### Author Response · Authors · 2025-11-27
> > > > **Author Response to Reviewer**
> > > >
> > > > Thank you for the follow-up. We’re glad our rebuttal helped address your concerns and we appreciate your time and consideration throughout the review process.

---

### Official Review · Reviewer_zkoA · 2025-11-04

**Soundness:** 4
**Presentation:** 4
**Contribution:** 3
**Rating:** 6
**Confidence:** 3

**Summary:**

The paper introduces TGM, a modular and efficient open-source library for temporal graph learning.
It unifies continuous-time and discrete-time dynamic graph paradigms within a single framework, providing native support for time operations, event-driven iteration, and node/edge dynamics.
TGM uses a _hook_ abstraction to implement flexible composition of temporal graph operations and efficient workflows.
The experiments show convincing speedups on a broad set of models.

**Strengths:**

- Unification: Comprehensive integration of CTDG and DTDG models under a single abstraction.

- Technical design and efficiency: The hook-based modularity and vectorized implementation lead to significant computational gains.

- Empirical validation and usability: Extensive benchmarks on multiple datasets and models.

**Weaknesses:**

-Limited novelty beyond software engineering: While the framework is well-engineered, the conceptual contribution (e.g., the hook abstraction) is mostly organizational rather than methodological.

- Evaluation scope: The Experiments focus mainly on efficiency and reproducibility; some examples showing directions of possible novel research would strengthen the claim of scientific impact.

**Questions:**

Apart from the points discussed above, there are the following minor points:

- Could the authors clarify how hooks differ from standard PyTorch data transformations or DGL message-passing pipelines conceptually?

- The paper mentions “dynamic graph property prediction” as a novel task; could more examples or datasets be shown?

- How does discretization handle overlapping time intervals or missing timestamps in real-world data? Or edges appearing and disappearing within the same time interval.

- Are there guidelines for selecting optimal time granularities, given their strong impact on performance?

- Representing Continuous-Time and Discrete-Time Graphs: A similar idea has been used already in [1], Definition 2 and 3.

[1] A. Longa et al., Graph Neural Networks for Temporal Graphs: State of the Art, Open Challenges, and Opportunities, TMLR (2023)

---

> ### Author Response · Authors · 2025-11-22
> **Author Response to Reviewer zkoA (Part 1)**
>
> We thank the reviewer for taking the time to review our paper and for offering valuable suggestions. We are pleased that the reviewer recognizes the importance of our work in providing a single abstraction for both CTDG and DTDG models; significant computational gains of TGM’s hook-based modularity and vectorized implementation. Here, we address each point raised by the reviewer as follows and revise our manuscript to incorporate the reviewer’s feedback in blue in the updated PDF.
>
> ---
>
> **W1. Novelty beyond software engineering**
>
> > W1 Limited novelty beyond software engineering: While the framework is well-engineered, the conceptual contribution (e.g., the hook abstraction) is mostly organizational rather than methodological.
>
> **Response:** Thank you for acknowledging the sound engineering of TGM. While the framework in TGM does provide significant organizational benefits, we respectfully argue that TGM has significant conceptual contributions that are highly impactful for the field of temporal graph learning as seen below.
>
> - **Methodological contribution.**
> TGM is the first framework to natively support both continuous-time and discrete-time temporal graph representations under a single abstraction. Our event-based and time-based iteration modes (Section 3) provide a standardized formulation that applies to both CTDG and DTDG models. In practice, choosing between CTDG and DTDG is non-trivial, and existing libraries force this choice upfront. For academic researchers, TGM offers an exciting opportunity to cross pollinate ideas between both types of methods while pushing the state-of-the-art.
>
> - **Many firsts of TGM.**
> TGM is the first to natively support dynamic node events expose time-granularity as a first-class concept with build-in conversion operators support link, node and graph-level dynamic tasks within a single framework
>
> - **Hook formalism.** TGM introduces the hook formalism as a powerful new abstraction for temporal graphs which standardize essential operations (e.g. neighbour sampling, negative sampling) while preventing temporal leakage. As shown in Figure 3, hooks also generalize to analytical workflows such as density-of-states or spectral statistics. This abstraction is more than organizational: it enables modular, composable pipelines that substantially accelerate research iteration.
>
> - **Standardization benefits for the community.**
> By unifying common but error-prone components (e.g. sampling, subgraph queries, temporal iteration), TGM provides robust infrastructure that allows new methods to be implemented by swapping modules (e.g., changing snapshot granularity in a single line). This lowers barriers and promotes reproducibility across the TGL community.
>
> Taken together, these contributions extend beyond software engineering: they provide the first conceptual and practical unification of temporal graph modeling, providing clear benefits to the TGL community.
>
> **W2. Evaluation scope**
>
> > Evaluation scope: The Experiments focus mainly on efficiency and reproducibility; some examples showing directions of possible novel research
>
> **Response:**
> We thank the reviewer for highlighting this point. We agree that demonstrating research possibilities is important, and we would like to respectfully note that Section 5.2 already presents three concrete research directions uniquely enabled by TGM’s unified design (all of which are not possible in existing libraries)
>
> - **RQ1: Graph Property Prediction via Time-Based Iteration**
> TGM’s novel iteration-by-time mechanism treats each time interval as a graph snapshot, enabling tasks such as forecasting future graph properties. This makes it easy for researchers to study questions like cross-task transfer (e.g., how a link-prediction trained model generalizes to node- or graph-level properties), which were previously cumbersome.
>
> - **RQ2: Effect of Time Granularity for DTDG Methods**
> Because TGM supports arbitrary time granularities with built-in conversion, we can directly test how real-world time resolution affects model performance. In Section 5.2, we show that varying granularity can change GCN performance by up to 30% on link prediction, suggesting a new research direction on learning or selecting optimal time resolutions.
>
> - **RQ3: Batch-size effects in CTDG Evaluation**
> TGM enables CTDG models to be evaluated both by event count and by snapshot. This provides the first platform for systematically studying how CTDG internal state evolution interacts with batch size or snapshot boundaries, opening new questions about how CTDG methods can be leveraged more effectively on snapshots rather than the event streams.

---

> > ### Author Response · Authors · 2025-11-22
> > **Author Response to Reviewer zkoA (Part 2)**
> >
> > **Q1. TGM hooks**
> >
> > > Could the authors clarify how hooks differ from standard PyTorch data transformations or DGL message-passing pipelines conceptually?
> >
> > **Response:**
> > We thank the reviewer for this interesting discussion. Conceptually, TGM hooks differ from PyTorch transforms and DGL message passing in three fundamental ways:
> >
> > 1. Hooks can be stateful, which is essential for temporal graphs
> > Temporal operations (e.g., neighbor sampling or negative sampling) depend on past events. In TGM, stateful hooks maintain and update internal state as new events arrive, and this state is correctly carried into validation and test (mirroring what the model is allowed to “remember”). PyTorch transforms, in contrast, are stateless and designed primarily for vision-style preprocessing.
> >
> > 2. Hooks expose a typed tensor I/O interface
> > TGM ensures composability by enforcing type-checked inputs and outputs across sequences of potentially complex operations. This enables safe construction of multi-step temporal pipelines, which goes beyond standard preprocessing APIs. It also unlocks the ability to efficiently schedule the required hook operations based on their dependencies.
> >
> > 3. Hooks are model-agnostic, and more general than message passing
> > DGL’s primitives (message, reduce, update) are tied to GNN computation. TGM hooks operate over the temporal graph itself and can express tasks far beyond GNN layers such as analytical pipelines (Figure 3). They structure temporal workflows, not just neural message passing.
> >
> > **Q2. Dynamic graph property prediction**
> >
> > > The paper mentions “dynamic graph property prediction” as a novel task; could more examples or datasets be shown?
> >
> > **Response:**  Thank you for this question. TGM is designed to make dynamic graph property prediction easy to instantiate on any temporal dataset, and we have expanded our experiments to demonstrate this. To provide additional evidence and datasets, we added native I/O support for MiNT [1], a recent benchmark specifically targeting dynamic graph property prediction. Using TGM, we evaluated three token-network datasets (ADX, ARC, MIR) on the MiNT task of predicting whether the number of transactions in a network will increase, a problem of practical importance in blockchain analytics [2].The results are summarized in the table below.
> >
> > ### Binary classification task — AUC (↑). Best in **bold** and second best in *italic*.
> >
> > | Model  | ADX | ARC | MIR |
> > |--------|-----|-----|-----|
> > | P.F.   | 0.397 ± 0.010 | _0.625 ± 0.022_ | 0.365 ± 0.011 |
> > | T-GCN  | **0.897 ± 0.019** | **0.901 ± 0.020** | **0.900 ± 0.008** |
> > | GCLSTM | _0.588 ± 0.047_ | 0.466 ± 0.019 | _0.656 ± 0.046_ |
> > | GCN    | 0.503 ± 0.049 | 0.608 ± 0.061 | 0.555 ± 0.033 |
> >
> > The table shows that T-GCN consistently outperforms all other models across the three datasets (ADX, ARC, MIR), achieving the highest mean scores with relatively low variability. GCLSTM and P.F. show competitive performance in some datasets, but their results are less stable, indicated by larger standard deviations. Overall, T-GCN demonstrates both superior accuracy in graph property prediction.
> >
> > [1] Shamsi K., et al. "MiNT: Multi-Network Transfer Benchmark for Temporal Graph Learning." NeurIPS 2025.
> >
> > [2] Akcora, C. G., et al. "Bitcoinheist: Topological data analysis for ransomware detection on the bitcoin blockchain."
> >
> > **Q3. Overlapping time intervals or missing timestamps**
> >
> > > How does discretization handle overlapping time intervals or missing timestamps in real-world data? Or edges appearing and disappearing within the same time interval.
> >
> > **Response:**
> > Thank you for the question. TGM’s time discretization is designed to be flexible while maintaining a clear and predictable temporal semantics.
> >
> > - **Overlapping time intervals**
> >  TGM’s current dataloader assumes non-overlapping, equal-sized intervals and processes them chronologically. If overlapping intervals are desired, TGM currently supports this via its graph API: users can provide an explicit list of start/end boundaries, allowing arbitrary (including overlapping) time window extraction. This can be incorporated without modifying model code.
> >
> > - **Missing or incomplete timestamps**
> > When the input lacks real timestamps TGM assigns an ‘ordered’ time granularity, meaning only event order is known. In this case, TGM supports iteration-by-event but not iteration-by-time (since no temporal scale exists). This avoids introducing artificial or misleading time intervals.
> >
> > - **Edges appearing and disappearing within the same interval**
> > Under discretization, edges are treated as spontaneous within the interval in which they occur. If a researcher needs to preserve finer temporal distinctions (e.g., multiple changes within one interval), we suggest choosing a finer time granularity so that each appearance/disappearance is captured at its original resolution.

---

> > > ### Author Response · Authors · 2025-11-22
> > > **Author Response to Reviewer zkoA (Part 3)**
> > >
> > > **Q4.Optimal time granularities**
> > >
> > > > Are there guidelines for selecting optimal time granularities, given their strong impact on performance?
> > >
> > > **Response:**
> > > Thank you for the question. One of TGM’s key motivations is precisely to enable systematic study of time-granularity selection, which has been difficult or impossible in prior libraries. We agree that this is an important research direction, and TGM provides the infrastructure to explore it rigorously.
> > > At present, we offer one practical guideline. For snapshot-based models (e.g., GCN, GCLSTM), choose a granularity that avoids empty snapshots. These models assume that each snapshot represents a meaningful, fixed-lag prediction target. If a chosen granularity produces intervals with no events, this assumption breaks, and performance can degrade. Beyond this heuristic, we expect that principled granularity selection (potentially data-driven or adaptive) will emerge as an important line of work enabled by TGM’s unified temporal graph APIs.
> > >
> > > **Q5.Representing CTDGs and DTDGs**
> > >
> > > > Representing Continuous-Time and Discrete-Time Graphs: A similar idea has been used already in [1], Definition 2 and 3.
> > >
> > > **Response:**
> > > Thank you for the comment. We agree that prior surveys (e.g., [1], [2]) have described continuous-time (CTDG) and discrete-time (DTDG) temporal graphs as two major methodological families. However, these works do not provide a unified formulation or tooling for treating them as compatible representations of the same underlying temporal graph.
> > >
> > > In the existing literature, CTDG and DTDG models are developed in isolation, with incompatible data formats, dataloaders, and evaluation pipelines making cross-comparison or integration difficult. TGM is the first TGL library to natively support both CTDG and DTDG within a single abstraction. Our key contribution is to reinterpret CTDG and DTDG simply as two iteration modes over the same temporal data. This is enabled by TGM’s novel notions of time granularity and time operations, which remove the structural friction between the two modalities.
> > >
> > > This unification allows researchers, for the first time, to:
> > > - load data once and benchmark CTDG and DTDG models interchangeably;
> > > - explore hybrid or adaptive representations;
> > > - directly compare methodological behavior under different granularities.
> > >
> > > Thus, while the categories themselves are known, TGM provides the first practical and conceptual unification that makes joint experimentation possible within one coherent library and API.
> > >
> > > [1] Longa et al., Graph Neural Networks for Temporal Graphs, TMLR 2023.
> > >
> > > [2] Kazemi et al., Representation Learning for Dynamic Graphs: A Survey, JMLR 2020.

---

> > > > ### Comment · Reviewer_zkoA · 2025-11-26
> > > > **Response to the Authors**
> > > >
> > > > I thank the authors for the detailed rebuttal and the updates to the manuscript. The clarifications strengthen the presentation and address my questions.

---

> > > > > ### Author Response · Authors · 2025-11-27
> > > > > **Author Response to Reviewer zkoA**
> > > > >
> > > > > We thank the reviewer for the positive follow-up and are glad our clarifications strengthen the presentation of the manuscript. Since the earlier concerns have been addressed, we kindly hope the reviewer may consider reflecting this in the final score. We appreciate the reviewer’s time and constructive feedback.

---

### Author Response · Authors · 2025-11-28
**Author Response to all Reviewers**

We thank the reviewers for their thoughtful reviews and positive reception of our work. Reviewers (KuR4 and pG5Q) mentioned that our work is a timely and valuable contribution to the field. Reviewers (zkoA and SxD6) highlighted the unification of CTDG and DTDG models in TGM. Reviewers (zkoA, SxD6 and pG5Q) acknowledged the efficiency and technical design of TGM while recognizing its hook-based modularity. We are encouraged to see that reviewer KuR4 mentioned that TGM could serve as a standard reference library for temporal graph learning research.

Indeed, our goal is to build TGM as the foundation and accelerator of future temporal graph learning research with its efficiency, diverse coverage, modular design and research flexibility. We appreciate the constructive feedback from reviewers on improving our paper and we have revised the manuscript based on suggestions from reviewers (the revised changes are marked in blue). We will continue to maintain and extend TGM with community feedback.

Here we would like to highlight changes and clarifications discussed in the rebuttal phase:

**Contributions of TGM**: TGM is the first temporal graph library to:
- Natively support dynamic node events.
- Treat time granularity and time conversions as first-class operators, enabling fine-grained temporal control and slicing.
- Unify link, node, and graph-level tasks within a single coherent framework.
- Conceptually unify CTDG and DTDG models via different modes of graph iteration.
- Introduce the hook formalism, a new abstraction that standardizes core TGL workflows (e.g., neighbor sampling, negative sampling, analytics).
- Provide robust and efficient infrastructure, achieving 7.8× average end-to-end training speedups over DyGLib and 175× average speedups on graph discretization compared to existing implementations.
- Offer the widest model coverage, including message-passing, transformer-based, and frontier architectures across link, node, and graph tasks.
- Enables rapid experimentation via its modular system design, facilitating extensions to broader datasets and tasks such as PyG Temporal and TGB-Seq.

**Inclusion of PyG Temporal Examples**. We added additional comparison with the PyG Temporal library for DTDG models. While TGM is not designed specifically for spatio-temporal forecasting, its flexible design allows us to integrate and run PyG Temporal tasks within TGM. We include examples on METR-LA and PEMS-BAY datasets, where our T-GCN implementation matches the reported performance of PyG Temporal while achieving similar efficiency. This demonstrates the generality and strength of TGM’s system design.

**Roadmap for next TGM release**:  Based on reviewer suggestions, we will support new models in TGM including EvolveGCNO, ROLAND and CTAN (increasing the model coverage to 11 models) as well as native IO support for TGB-seq (increasing the out-of-the-box dataset coverage to 24 diverse datasets) in the next TGM release. All of the above have been implemented in pull requests, and we will add full test coverage of these new models and datasets prior to the next TGM release.

**Additional graph property prediction datasets and experiments.**  To include more datasets and examples for graph property prediction as requested, we have added I/O support for MiNT, a recent benchmark on this task. We then ran TGM experiments on three token networks, ADX, ARC, and MIR, on the task of predicting whether the number of transactions in a token network will increase, a critical task in blockchain analysis. These results are now included in Table 8 of the revised paper.

---

### Author Response · Authors · 2025-11-30
**Summary of Rebuttal Changes and Discussions**

Dear Area Chair,

We sincerely appreciate the time and effort you are investing in reviewing our submission, especially given these challenging circumstances. To support your evaluation, we have included below a brief overview of the main changes made during the rebuttal period:

- **Reviewer zkoA (Score: 6):** expanded TGM support of graph property prediction task to MiNT [1] datasets, showing results of DTDG models on three new token networks. We clarified conceptual contributions in TGM and answered questions regarding TGM hooks and time granularities. The reviewer responded with “*The clarifications strengthen the presentation and address my questions.*”

- **Reviewer SxD6 (Score: 6):** added native support for spatio-temporal graph datasets to TGM, demonstrated matching efficiency and correctness on Pems-Bay and Metr-LA datasets to the original PyG Temporal implementation [2], showing the research flexibility of TGM. We clarified conceptual contributions in TGM and moved the correctness results to the revised main paper. We answered questions about batch processing, deletion events and irregularly sampled snapshots. The reviewer responded with “*Your reply has solved the majority of my problems*”.

- **Reviewer KuR4 (Score: 4 -> 6):** added experiments to demonstrate matching efficiency and correctness in TGM on Pems-Bay and Metr-LA datasets with that of the original PyG Temporal implementations. We clarified our experimental setup, parallel event processing and added discussion of batch-size tradeoffs in the appendix. The reviewer raised their score from 4 to 6 and mentioned that the authors “*clarified most of my concerns*” and is “*willing to further increase it if authors can strengthen the coverage of TGM*”. In light of this, we added TGM coverage to a new and widely-used method, ROLAND [3], reported its performance on tgbl-wiki and committed to adding two methods, EvolveGCNO [4] and CTAN [5], in the next TGM release.

- **Reviewer pG5Q (Score: 6):** added native IO support for TGB-Seq [6] datasets in TGM, and reported TGN and TGAT end-to-end latency on three datasets. We revised Section 3 for clarity, describing the core factors behind TGM’s efficiency. On Nov. 26th, the reviewer replied with “*it addresses most of my concerns*” and had one remaining question about TGM caching’s CPU memory. In response to this, we added an experiment showing that TGM’s CPU memory overhead is substantially lower than DyGLib’s and clarified our caching strategy.

These changes led to **scores of 6, 6, 6 and 6**, prior to the freezing of the review period. Additional details are available in the author's response to all reviewers comment. We hope this summary is beneficial for your assessment.

Best,

Authors of Submission 9740

---

[1] Shamsi, Kiarash, et al. "MiNT: Multi-Network Transfer Benchmark for Temporal Graph Learning." The Thirty-ninth Annual Conference on Neural Information Processing Systems Datasets and Benchmarks Track.

[2] Rozemberczki, Benedek, et al. "Pytorch geometric temporal: Spatiotemporal signal processing with neural machine learning models." Proceedings of the 30th ACM international conference on information & knowledge management. 2021.

[3] You, Jiaxuan, Tianyu Du, and Jure Leskovec. "ROLAND: graph learning framework for dynamic graphs." Proceedings of the 28th ACM SIGKDD conference on knowledge discovery and data mining. 2022.

[4] Pareja, Aldo, et al. "Evolvegcn: Evolving graph convolutional networks for dynamic graphs." Proceedings of the AAAI conference on artificial intelligence. Vol. 34. No. 04. 2020.

[5] Gravina, Alessio, et al. "Long Range Propagation on Continuous-Time Dynamic Graphs." International Conference on Machine Learning. PMLR, 2024.

[6] Yi, Lu, et al. "TGB-Seq Benchmark: Challenging Temporal GNNs with Complex Sequential Dynamics." The Thirteenth International Conference on Learning Representations.

---

### Meta-Review · Area_Chair_7wU3 · 2026-01-01

**Summary:**

This work proposes TGM, a research-oriented library for machine learning on temporal graphs. TGM unifies CTDG and DTDG approaches, offers first-class support for dynamic node features, time-granularity conversions, and native handling of link-, node-, and graph-level tasks. Empirically, it achieves significant speedups over prior relevant libraries.
During the review process, the following concerns were raised:

* Limited novelty beyond software engineering


* Limited evaluation scope


* Limited DTDG methods and experiments


* Concerns regarding official implementation correctness


* Lack of empirical comparison with other libraries


* Missing important implementation details (e.g., data splits, hyperparameters, handling deletion events, and how batches are processed)


* Known models and datasets missing from TGM

**Reviewer Concerns:**

* In the rebuttal, the authors claim that TGM unifies key concepts in temporal graph learning (CTDG vs. DTDG, and node-, link-, and graph-level tasks). They also claim to standardize temporal graph operations via appropriate abstractions. In my view, the authors adequately demonstrate novelty beyond software engineering.

* The authors also clarify in the rebuttal that additional experiments are already included in the manuscript (e.g., Subsection 5.2 and Appendix B.4), which addresses the limited evaluation scope concern.

* However, the authors did not add further DTDG experiments or additional DTDG methods in the rebuttal, stating instead that they plan to do so in the future. Therefore, this concern remains unresolved.

* Regarding implementation correctness, the authors state in the rebuttal that experiments run using TGM reproduce the same results as those obtained via the TGB evaluation procedure, and the codebase appears to be extensively tested. Thus, this concern is largely resolved.

* The authors added a training-time comparison with PyG Temporal in the rebuttal, which addresses the lack of empirical comparison with other libraries.

* They also provided additional implementation details in the rebuttal (including splits and hyperparameters, as well as clarifications around deletion events and batching).

* Finally, the authors did not incorporate the additional models and datasets suggested by reviewers.

**Reviewer Scores:**

Based on the discussion log, Reviewers zkoA, SxD6, and pG5Q participated in the discussion, stated that most of their concerns were resolved, and kept their scores at 6. Reviewer KuR4 also participated and increased their score from 4 to 6. By the end of the discussion, all reviewers expressed a positive stance toward acceptance.

---

### Decision · Program_Chairs · 2026-01-26

Accept (Poster)